# Therapeutic deep brain stimulation disrupts movement-related subthalamic nucleus activity in parkinsonian mice

Jonathan S Schor[1,2,3], Isabelle Gonzalez Montalvo[1,2,3], Perry WE Spratt[1,2,3], Rea J Brakaj[2,3,4], Jasmine A Stansil[2,3,4,5], Emily L Twedell[1,2,3,5], Kevin J Bender[1,2,3,4], Alexandra B Nelson[1,2,3,4,5]*

[1]Neuroscience Program, University of California, San Francisco, San Francisco, United States; [2]Kavli Institute for Fundamental Neuroscience, University of California, San Francisco, United States; [3]Weill Institute for Neuroscience, University of California,, San Francisco, United States; [4]Department of Neurology, University of California, San Francisco, San Francisco, United States; [5]Aligning Science Across Parkinson's (ASAP) Collaborative Research Network, Chevy Chase, United States

*For correspondence:
Alexandra.Nelson@ucsf.edu

Competing interest: The authors declare that no competing interests exist.

## Abstract
Subthalamic nucleus deep brain stimulation (STN DBS) relieves many motor symptoms of Parkinson's disease (PD), but its underlying therapeutic mechanisms remain unclear. Since its advent, three major theories have been proposed: (1) DBS inhibits the STN and basal ganglia output; (2) DBS antidromically activates motor cortex; and (3) DBS disrupts firing dynamics within the STN. Previously, stimulation-related electrical artifacts limited mechanistic investigations using electrophysiology. We used electrical artifact-free GCaMP fiber photometry to investigate activity in basal ganglia nuclei during STN DBS in parkinsonian mice. To test whether the observed changes in activity were sufficient to relieve motor symptoms, we then combined electrophysiological recording with targeted optical DBS protocols. Our findings suggest that STN DBS exerts its therapeutic effect through the disruption of movement-related STN activity, rather than inhibition or antidromic activation. These results provide insight into optimizing PD treatments and establish an approach for investigating DBS in other neuropsychiatric conditions.

## Editor's evaluation

Using a combination of electrical artifact-free calcium imaging and electrical stimulation, the authors probe the effects of stimulation on the neural dynamics of basal ganglia structures that correlate with motor improvement. The paper would be of interest to neuroscientists and clinician scientists interested in better understanding the mechanism of deep brain stimulation (DBS) in the treatment of Parkinson's disease.

## Introduction

The basal ganglia are a group of interconnected subcortical structures long believed to control movement through modulation of neuronal firing. This rate-based model posits that reductions in basal ganglia output (globus pallidus interna [GPi] and substantia nigra pars reticulata [SNr]) facilitate movement. According to this model, the loss of dopaminergic input to the basal ganglia in Parkinson's disease (PD) leads to pathological increases in GPi/SNr firing and impaired movement. Dopamine replacement therapies, such as with dopamine agonists or the dopamine precursor levodopa, reduce GPi/SNr firing in both PD patients and animal models (*Hutchinson et al., 1997*; *Levy et al., 2001*;

*Lozano et al., 2000*; *Papa et al., 1999*), suggesting they may act by restoring normal firing rates. Given these findings and the rate model, it may seem paradoxical that deep brain stimulation (DBS) of the subthalamic nucleus (STN), an intrinsic basal ganglia nucleus with predominantly excitatory projections to GPi/SNr, is one of the most effective treatments for PD (*Hickey and Stacy, 2016*).

Investigations into how STN DBS impacts STN, GPi, and SNr activity have yielded conflicting results, and thus how STN DBS exerts its therapeutic effects remains unclear. However, three major theories have been suggested (*Chiken and Nambu, 2014*): (1) STN DBS *inhibits* STN activity, consistent with the rate model; (2) STN DBS bypasses basal ganglia output by antidromically *exciting* cortical neurons projecting to the STN; or (3) STN DBS *disrupts* movement-related activity in the STN. Supporting (1), focal STN lesions relieve motor symptoms of PD (*Andy et al., 1963*; *Bergman et al., 1990*), and some groups have observed inhibition of STN, GPi, or SNr firing during DBS (*Filali et al., 2004*; *Moran et al., 2011*). However, other groups have observed excitation in these structures (*Hashimoto et al., 2003*; *Reese et al., 2011*). Supporting (2), antidromic activation of primary motor cortex (M1) during STN DBS has been observed in rodents and in humans (*Li et al., 2012*; *Miocinovic et al., 2018*), and optical stimulation of hyperdirect M1 neurons relieves motor symptoms in parkinsonian mice (*Gradinaru et al., 2009*; *Sanders and Jaeger, 2016*). However, recent evidence in nonhuman primates (NHPs) shows antidromic M1 activation during electrical STN DBS to be both variable and transient, suggesting this pathway may not be the primary mechanism of STN DBS (*Johnson et al., 2020*). Supporting (3), correlations have been found between parkinsonian motor symptoms and signals such as local field potential (LFP) oscillations (*Kühn et al., 2008*; *Stein and Bar-Gad, 2013*; *Wingeier et al., 2006*) and bursting (*Pan et al., 2016*; *Zhuang et al., 2018*). Despite these observations, it has been difficult to distinguish between these possible mechanisms using electrophysiology, or to causally link observed changes in pattern and rhythm with improvements in behavior, in part due to large DBS-related electrical artifacts in recordings near the DBS site.

Here, we used the genetically encoded calcium indicator GCaMP to enable region- and cell type-specific (and electrical artifact-free) optical recording of neural activity in three basal ganglia circuit nodes (STN, SNr, and M1) in a mouse model of STN DBS for PD. Surprisingly, we found that STN DBS *increased* activity in the STN and the SNr (in conflict with the rate model), and though DBS also altered hyperdirect pathway M1 activity, these changes did not correlate strongly with motor improvement. Furthermore, M1 lesions did not eliminate the therapeutic benefit of STN DBS. Finally, we found that DBS abolished stereotyped patterns of STN activity around movement onset. An optical stimulation protocol that similarly attenuated this activity pattern was sufficient to improve movement, suggesting that disruption of movement-related STN activity may be a core therapeutic mechanism of STN DBS. Together, our results suggest that STN DBS causes specific disruptions in motor signals at the level of the STN, broadening our understanding of how the basal ganglia mediates motor control.

## Results

### STN GCaMP signals correlate with spiking measured by electrophysiology

To test whether DBS inhibits, excites, or disrupts its target neurons, we used fiber photometry to measure bulk fluorescence signals from the genetically encoded calcium indicator GCaMP6s. To validate this approach, we first sought to determine whether STN calcium signals could serve as proxy for neural activity. We injected the STN of VGlut2-Cre mice with AAVs encoding Cre-dependent GCaMP6s, limiting expression to glutamatergic STN neurons. We then performed simultaneous whole-cell current-clamp recordings and fluorescence imaging of STN neurons in ex vivo slices to compare firing rate and intracellular calcium signals (*Figure 1A–C*). Neurons were driven to spike with intracellular current pulses at frequencies ranging from 10 to 200 Hz in 1 min epochs, which evoked rhythmic spiking at these frequencies (*Figure 1D–E*). Calcium, as measured by changes in GCaMP6s fluorescence, similarly increased during firing at 10–200 Hz (*Figure 1F–G*). In a subset of recordings, we assessed GCaMP6s signals in response to a range of lower firing frequencies (10–60 Hz) or higher frequencies (up to 200 Hz). Again, STN neurons showed rhythmic spiking that matched the frequency of pulsatile stimulation (*Figure 1E*, insets), and calcium signals correlated with firing rates up to about 100 Hz, and were variable between 100 and 200 Hz (*Figure 1G*, inset). As some groups have speculated that high-frequency DBS may cause some neurons to enter depolarization block, we

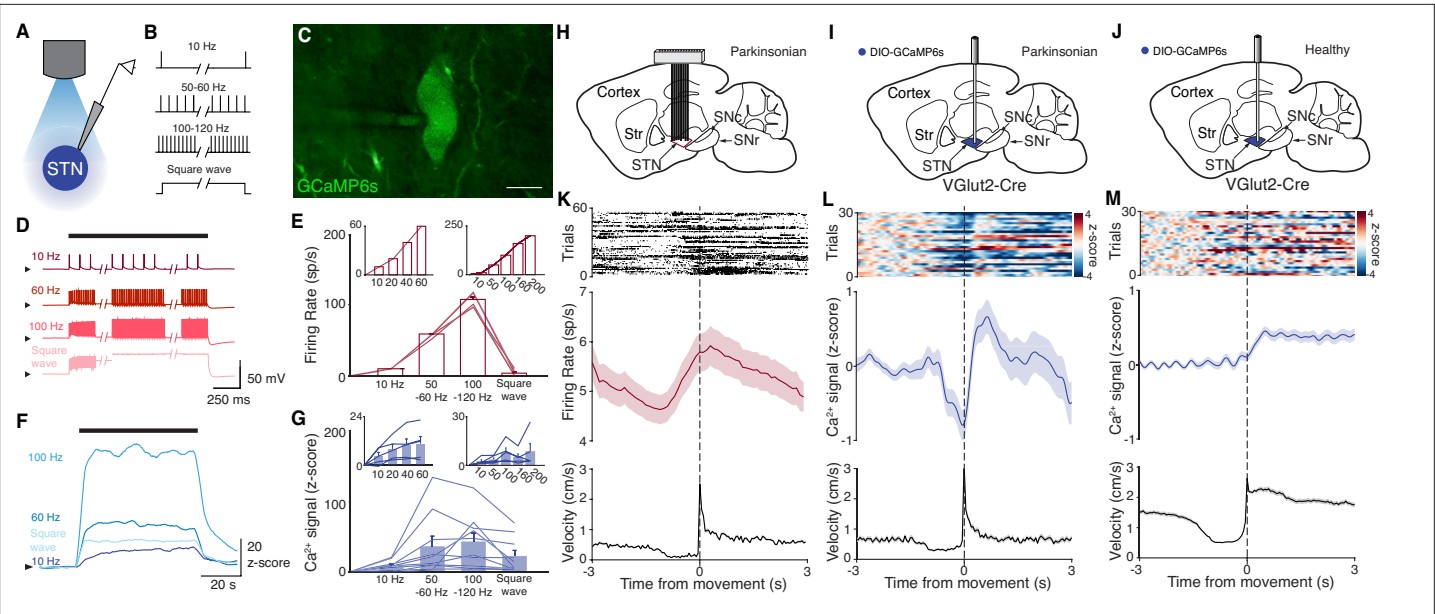

**Figure 1.** Subthalamic nucleus (STN) GCaMP signals correlate with spiking measured by electrophysiology. (**A–G**) Combined ex vivo electrophysiological and calcium imaging recordings in STN neurons. Neurons were patched in the whole-cell current-clamp configuration. (**A**) Combined whole-cell/calcium imaging recording configuration. (**B**) Brief intracellular current pulses were provided at different frequencies up to 200 Hz and with prolonged depolarization ('square wave'). (**C**) Image of GCaMP-expressing STN neuron (scale bar = 10 μm). (**D**) Representative STN neuron responses to intracellular current pulses at different frequencies and to prolonged depolarization ('square wave'). (**E**) Average firing rate of STN neurons in response to stimulation at a range of frequencies and prolonged depolarization (n=10 cells, N=3 mice for main; n=5 cells, N=1 mouse for each inset). (**F**) Representative trace of z-scored STN GCaMP signal in response to current-clamp stimulation. (**G**) Average z-scored STN GCaMP signal in response to intracellular current pulses at a range of frequencies and prolonged depolarization (n=10 cells, N=3 mice for main; n=5 cells, N=1 mouse for each inset). Arrowhead in current-clamp traces and GCaMP traces corresponds to –75 mV and 0 z-score, respectively. Bar plots show mean ± SEM. (**H–M**) In vivo electrophysiological and GCaMP fiber photometry recordings in STN neurons from freely moving mice, aligned to movement starts. (**H**) Sagittal schematic showing multielectrode array implant in the STN or parkinsonian mice for single-unit electrophysiology. (**I, J**) Sagittal schematic showing STN GCaMP and fiber implant for photometry in parkinsonian (**I**) and healthy (**J**) mice. (**K**) Representative STN single-unit firing (top), average firing rate (middle), and average velocity (bottom) aligned to movement starts (n=17 cells, N=3 mice). (**L, M**) Representative STN fiber photometry signal (top), average fiber photometry signal (middle), and average velocity (bottom) aligned to movement starts in parkinsonian (L, N=8) and healthy (M, N=5) mice. Average firing rate, photometry, and velocity traces show mean ± SEM.

The online version of this article includes the following figure supplement(s) for figure 1:

**Figure supplement 1.** Hemiparkinsonian mice show decreased movement velocity and ipsilesional rotation bias.

also tested the firing and calcium signals associated with constant current ('square wave') stimulation. During such stimulation, STN neurons fired only transiently, appearing to enter depolarization block (*Figure 1D–E*). Under these circumstances, evoked calcium signals fell between those associated with 10 and 50–60 Hz spiking (*Figure 1F–G*). These experiments suggest that the relationship between spiking and GCaMP calcium signals may break down at very high firing frequencies, or under conditions of forced depolarization block. However, at more moderate firing frequencies, GCaMP calcium signals correlate with STN firing.

We next tested whether changes in bulk GCaMP6s fluorescence, as recorded through in vivo fiber photometry, corresponded with in vivo single-unit activity. To address this question, we recorded neural activity in vivo in two sets of parkinsonian animals, using either electrophysiology or GCaMP fiber photometry. We rendered mice parkinsonian through unilateral injection of 6-hydroxydopamine (6-OHDA) in the medial forebrain bundle (MFB, *Figure 1—figure supplement 1A*). In one group of parkinsonian mice, we implanted a 16-channel electrode array in the ipsilateral STN (*Figure 1H*). In a second group of parkinsonian VGlut2-Cre mice, we injected Cre-dependent GCaMP6s and implanted an optical fiber in the ipsilateral STN (*Figure 1I*). As observed in prior studies (*Bové and Perier, 2012*; *Campos et al., 2013*; *Carvalho et al., 2013*; *Ungerstedt, 1968*), 6-OHDA-treated mice showed both decreased movement velocity (1.3±0.1 cm/s parkinsonian vs. 3.1±0.4 cm/s healthy, p=8.23 × 10⁻⁵) and an ipsilesional rotational bias (p=1.40 × 10⁻³) when compared to healthy mice (*Figure 1—figure*

supplement 1B). Additionally, movement velocity and rotation bias did not differ significantly between parkinsonian mice with and without STN implants (*Figure 1—figure supplement 1B*, p=0.66 for velocity, p=0.97 for rotation), suggesting that local, implant-related STN tissue disturbance did not alter gross movement parameters. We then aligned single-unit spiking activity and fiber photometry signal to movement starts (defined as a transition from velocity <0.5 to >2 cm/s) in both sets of mice (*Figure 1K–L*). STN single units showed baseline firing rates of approximately 5–10 spikes/s (*Figure 1K*), well within the linear range for GCaMP6s (*Figure 1G*, inset). The firing of STN units showed a marked change in firing rate around movement onset, increasing just prior to, and peaking just following movement initiation (approximately 1 spike/s over the baseline rate). Population calcium signals showed a similar increase around movement onset, but lagged the rise in single-unit firing rate by ~1 s, likely due to the slower kinetics of GCaMP6s (*Markowitz et al., 2018*; *Figure 1L*). To determine whether this movement-related activity was also present in the STN of healthy animals, we used a similar fiber photometry approach, targeting GCaMP to STN glutamatergic neurons (*Figure 1J*). Healthy animals tended to move at greater average velocity, but using the same event selection criteria, we aligned STN calcium signals to movement onset. As in parkinsonian mice, we found that in healthy animals, STN GCaMP signals increased around movement onset (*Figure 1M*). Thus, STN calcium signals and electrophysiology appear to capture a slightly time-shifted, but qualitatively similar increase in activity when aligned to behavior. Together, these ex vivo and in vivo experiments show a correlation between spiking and GCaMP6s signal, and therefore support the utility of fiber photometry as a proxy for neuronal activity in the context of STN DBS.

## STN DBS consistently increases STN activity

The direct impact of STN DBS on STN neural activity remains unclear: some studies indicate that STN DBS decreases STN firing rates, while recordings in downstream nuclei indicate STN activity may increase. To address whether STN DBS increases or decreases overall STN activity, we injected parkinsonian VGlut2-Cre mice with Cre-dependent GCaMP6s and implanted them with both an STN DBS device and an optical fiber (*Figure 2A–B*; *Figure 2—figure supplements 1A and 2A*). We applied DBS using 12 different parameter sets, holding pulse width and amplitude constant (60 µs and 200 µA, respectively) and varying frequency (*Supplementary file 1*, table 2). Consistent with our previous findings in the same mouse model (*Schor and Nelson, 2019*), many different parameter sets were effective, across a broad frequency range. Grouping these parameter sets by stimulation frequency, we found that average movement velocity was increased by low- (5–40 Hz), medium- (60–100 Hz), and high-frequency (120–180 Hz) STN DBS (*Figure 2—figure supplement 2*; *Figure 3—figure supplement 1*; *Figure 4—figure supplement 1*). We found that 60–100 and 120–180 Hz DBS produced similar benefits in movement velocity, so focused on 60 and 100 Hz stimulation for subsequent analyses; 60 or 100 Hz stimulation improved multiple movement metrics (*Figure 2—figure supplement 1B-K*). At either 60 or 100 Hz stimulation, STN DBS increased movement velocity (*Figure 2—figure supplement 1B,C,G,H*; p=3.92 × 10$^{-9}$ for 60 Hz, p=5.53 × 10$^{-8}$ for 100 Hz) and percent time moving (*Figure 2—figure supplement 1D,I*; p=3.53 × 10$^{-9}$ for 60 Hz, p=1.46 × 10$^{-8}$ for 100 Hz), while not significantly altering rotation bias (*Figure 2—figure supplement 1E,J*; p=0.71 for 60 Hz, p=0.47 for 100 Hz) or causing prolonged dyskinesias (*Figure 2—figure supplement 1F,K*). We subsequently used movement velocity as the primary behavioral outcome measure for STN DBS. We then measured changes in STN activity in vivo in response to treatment with STN DBS. Surprisingly, in parallel with its impact on movement velocity (p=7.11 × 10$^{-9}$), STN DBS at 60 Hz caused a significant increase in STN calcium signals (*Figure 2C–D*, *Figure 2—figure supplement 2B*; p=1.19 × 10$^{-7}$). The same was true for stimulation at 100 Hz, with an increase in STN activity (p=9.54 × 10$^{-7}$) mirroring an increase in velocity (*Figure 2E–F*, *Figure 2—figure supplement 2C*; p=1.11 × 10$^{-9}$). A similar phenomenon could be seen with DBS parameters across a wide range of stimulation frequencies (*Figure 2—figure supplement 2D*, *Supplementary file 1*, table 2). This result suggests that, rather than inhibiting the STN, STN DBS increases STN activity.

These results appear to conflict with the proposed inhibitory mechanism of other PD treatments, such as surgical ablations and dopamine replacement therapy. To directly compare STN DBS and dopamine replacement therapy using the same readout of STN activity, we administered levodopa in the same set of mice. In these animals, levodopa increased movement velocity (*Figure 2—figure supplement 1L-M*; p=9.69 × 10$^{-10}$) and percent time moving (*Figure 2—figure supplement 1N*;

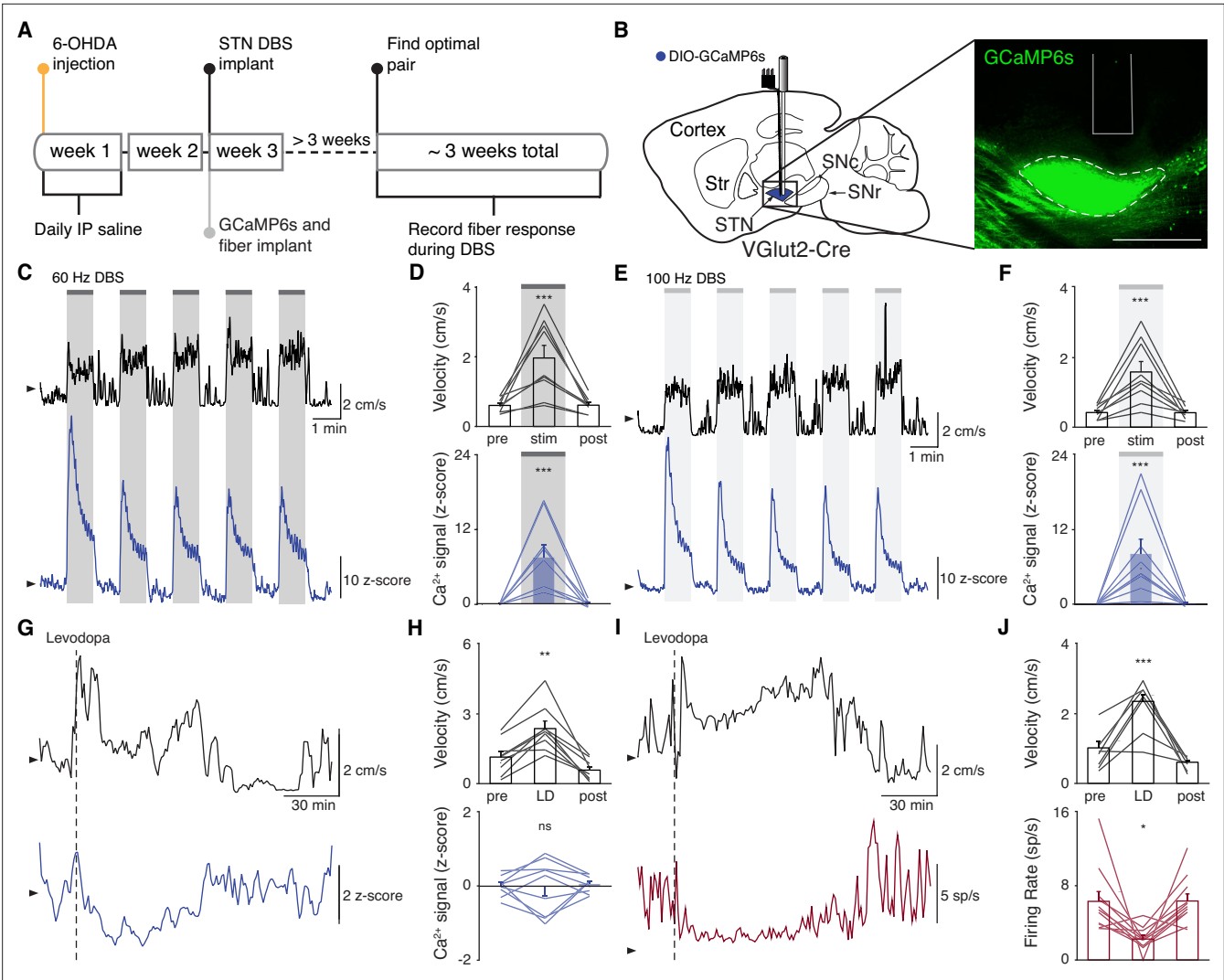

**Figure 2.** Subthalamic nucleus deep brain stimulation (STN DBS) consistently increases STN activity. (**A**) Experimental timeline. (**B**) Left: Sagittal schematic showing STN DBS and GCaMP fiber photometry. Right: Postmortem sagittal section showing GCaMP expression and estimated fiber placement in the STN (inset, scale = 500 µm). (**C**) Representative single-session velocity (black) and STN GCaMP signal (blue) in response to 60 Hz STN DBS. (**D**) Average velocity (top) and STN GCaMP signal (bottom) before, during, and after 60 Hz STN DBS (N=9 mice). (**E**) Representative single-session velocity (black) and STN GCaMP signal (blue) in response to 100 Hz STN DBS. (**F**) Average velocity (top) and STN GCaMP signal (bottom) before, during, and after 100 Hz STN DBS (N=9 mice). (**G**) Representative single-session velocity (black) and STN GCaMP signal (blue) before and after levodopa injection (dotted line). (**H**) Average velocity (top) and STN GCaMP signal (bottom) before, during, and after levodopa treatment (N=9 mice). (**I**) Representative single-session velocity (black) and STN single-unit activity (red) before and after levodopa injection (dotted line). (**J**) Average velocity (top) and STN single-unit activity (bottom) before, during, and after levodopa treatment (n=11 cells, N=3 mice). Statistical significance was determined using a one-way repeated measures ANOVA with a Tukey HSD post hoc analysis applied to correct for multiple comparisons; *p < 0.05, **p < 0.01, ***p < 0.001 (only comparison between pre and stim/LD shown, see *Supplementary file 1*, table 1 for detailed statistics). Arrowhead in velocity, GCaMP, and single-unit electrophysiology traces corresponds to 1 cm/s, 0 z-score, and 0 spike/s, respectively. Bar plots show mean ± SEM.

The online version of this article includes the following figure supplement(s) for figure 2:

**Figure supplement 1.** STN DBS and levodopa alleviate some motor signs in parkinsonian mice.

**Figure supplement 2.** STN DBS increases STN activity in vivo.

p=9.56 × 10$^{-10}$), evoked contralesional rotations (*Figure 2—figure supplement 1O*; p=1.73 × 10$^{-6}$), and caused minimal dyskinesias (*Figure 2—figure supplement 1P*). We subsequently used movement velocity and rotation bias as primary and secondary behavioral outcome measures of levodopa treatment, respectively. In these sessions, though all mice showed improvements in movement (p=1.66 × 10$^{-3}$ for velocity, p=3.91 × 10$^{-3}$ for rotation bias), changes in STN calcium were variable (*Figure 2G–H*;

*Figure 2—figure supplement 2D*). In some mice, STN activity decreased, while in others it increased: STN activity was not significantly changed across the entire group (*Figure 2H*; p=0.90). Injection with saline did not improve movement parameters, nor did it produce significant changes in STN activity (*Figure 2—figure supplement 2E-F*; p=0.084). As levodopa does not produce electrical artifacts like DBS, we were able to directly compare its effects on calcium signals and the gold standard measure of neural activity, single-unit electrophysiology. In parallel with its impact on movement (p=3.13 × $10^{-4}$ for velocity, p=9.77 × $10^{-4}$ for rotation bias), levodopa caused a modest, though significant, decrease in STN firing rates (*Figure 2I–J*, *Figure 2—figure supplement 2G*; p=0.029). The fact that bulk calcium signals did not reflect the modest reductions in STN firing seen with electrophysiology may relate to differential sensitivity of the two methods. Despite the fact that DBS and levodopa both improve movement parameters, they appear to alter overall activity level in opposite directions.

## STN DBS increases SNr activity

Though the STN is a critical node within the basal ganglia circuit, especially in regard to dysfunction in PD and its treatment, it is believed to regulate motor function via excitatory projections to basal ganglia output nuclei. In addition, STN DBS is likely to cause changes in the activity of nearby axons, and thus may have complex downstream effects. Therefore, while we did not observe inhibition at the level of the STN during STN DBS, we wondered if it might still produce inhibition at the level of the primary basal ganglia output nucleus in rodents, the SNr. To determine how SNr activity responds to STN DBS, we injected either VGAT-Cre mice with Cre-dependent GCaMP6s in the SNr (N=6 mice) or wild-type (WT) mice with synapsin- GCaMP6s in the SNr (N=2 mice) and implanted them with an STN DBS device and an optical fiber in the SNr (*Figure 3A*, *Figure 3—figure supplement 1A*). As before, STN DBS in these mice increased movement velocity across a wide range of frequencies (*Figure 3B–E*; p=1.33 × $10^{-5}$ for 60 Hz, p=4.99 × $10^{-6}$ for 100 Hz; *Figure 3—figure supplement 1D*, *Supplementary file 1*, table 2). Consistent with our results in the STN, STN DBS (at both 60 and 100 Hz) increased SNr activity (*Figure 3B–E*; *Figure 3—figure supplement 1B-D*; p=4.14 × $10^{-5}$ for 60 Hz, p=3.32 × $10^{-5}$ for 100 Hz). Contrary to the basal ganglia rate model, and the inhibition theory of DBS, these findings suggest that both STN and SNr activities are increased by STN DBS.

To again validate GCaMP fiber photometry signals and compare DBS to other treatments, we measured how levodopa altered SNr activity. In the same parkinsonian mice, levodopa increased movement velocity (p=0.016) and caused a contralesional rotation bias (*Figure 3F–G*; *Figure 3—figure supplement 1E*; p=7.81 × $10^{-3}$). In parallel, we observed a marked decrease in SNr neural activity as measured by fiber photometry (*Figure 3F–G*; *Figure 3—figure supplement 1E*; p=6.68 × $10^{-6}$). In contrast, saline neither improved movement parameters nor significantly changed SNr calcium signals (*Figure 3—figure supplement 1F-G*; p=0.31). Single-unit electrophysiological recordings also showed profound reductions in SNr firing rate (p=0.012) during therapeutic levodopa treatment (*Figure 3H–I*; *Figure 3—figure supplement 1H*), similar to findings in the GPi of parkinsonian NHPs (*Papa et al., 1999*). Thus, GCaMP fiber photometry and electrophysiology revealed qualitatively similar changes in SNr activity in response to levodopa, supporting the idea that these two measures of neural activity have substantial concordance. Furthermore, these experiments show marked differences in how STN DBS and levodopa impact neural activity, suggesting that STN DBS does not exert therapeutic effects through inhibition of STN or SNr.

## Hyperdirect pathway activity during STN DBS

While we did not observe inhibition in either the STN or SNr during STN DBS, a more recent theory posits that STN DBS acts through antidromic activation of the hyperdirect pathway: primary motor cortex (M1) neurons that project monosynaptically to the STN. To assess whether STN DBS increases activity of hyperdirect M1 neurons, we imaged hyperdirect pathway neurons using a retrograde viral strategy. We injected the STN of parkinsonian mice with one of two retrograde viruses encoding Cre recombinase (CAV2-Cre or rAAV2-Cre-mCherry), and injected M1 with Cre-dependent GCaMP6s (*Figure 4A*, *Figure 4—figure supplement 1A-B*). This strategy restricted expression of GCaMP6s to STN-projecting M1 neurons, which previously have been shown to send collaterals to the STN with parent axons in the cerebral peduncle (*Kita and Kita, 2012*; *Figure 4—figure supplement 1B*). We then implanted an optical fiber in M1 and a DBS device in the STN. As in other parkinsonian mice, DBS increased movement velocity across a wide range of stimulation frequencies (*Figure 4—figure*

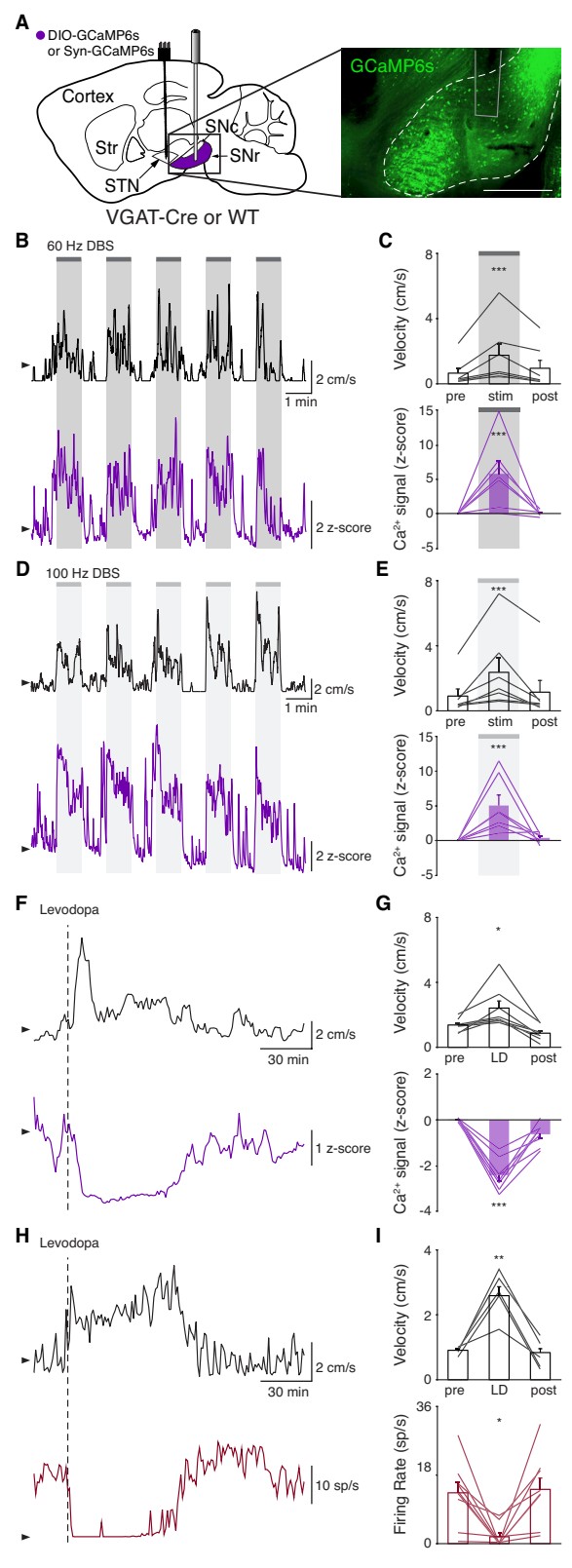

**Figure 3.** Subthalamic nucleus deep brain stimulation (STN DBS) increases basal ganglia output. (**A**) Left: Sagittal schematic showing STN DBS and substantia nigra pars reticulata (SNr) GCaMP fiber photometry. Right: Postmortem sagittal section showing GCaMP expression and estimated fiber placement in the SNr (inset, scale = 500 µm). (**B**) Representative single-session velocity (black) and SNr GCaMP signal (purple) in response to 60 Hz

*Figure 3 continued on next page*

*Figure 3 continued*

STN DBS. (**C**) Average velocity (top) and SNr GCaMP signal (bottom) before, during, and after 60 Hz STN DBS (N=7 mice). (**D**) Representative single-session velocity (black) and SNr GCaMP signal (purple) in response to 100 Hz STN DBS. (**E**) Average velocity (top) and SNr GCaMP signal (bottom) before, during, and after 100 Hz STN DBS (N=7 mice). (**F**) Representative single-session velocity (black) and SNr GCaMP signal (purple) before and after levodopa injection (dotted line). (**G**) Average velocity (top) and SNr GCaMP signal (bottom) before, during, and after levodopa treatment (N=8 mice). (**H**) Representative single-session velocity (black) and SNr single-unit activity (red) before and after levodopa injection (dotted line). (**I**) Average velocity (top) and SNr single-unit activity (bottom) before, during, and after levodopa treatment (n=9 cells, N=3 mice). Statistical significance was determined using a one-way repeated measures ANOVA with a Tukey HSD post hoc analysis applied to correct for multiple comparisons; *p < 0.05, **p < 0.01, ***p < 0.001 (only comparison between pre and stim/LD shown, see ***Supplementary file 1***, table 1 for detailed statistics). Arrowhead in velocity, GCaMP, and single-unit electrophysiology traces corresponds to 1 cm/s, 0 z-score, and 0 spike/s, respectively. Bar plots show mean ± SEM.

The online version of this article includes the following figure supplement(s) for figure 3:

**Figure supplement 1.** STN DBS evokes a rapid increase in SNr activity.

---

*supplement 1E, Supplementary file 1*, table 2; *Figure 4B–E*; p=1.51 × 10⁻⁹ for 60 Hz, p=4.14 × 10⁻⁷ for 100 Hz). Despite the consistent therapeutic effects of 60 Hz STN DBS (*Figure 4C*, top), hyperdirect M1 calcium responses were surprisingly variable: some mice showed increases, while the calcium signal in other mice decreased or did not change (*Figure 4C*, bottom; *Figure 4—figure supplement 1C*; p=0.084). In the same mice, 100 Hz STN DBS also increased movement velocity (*Figure 4E*, top), but in this case M1 activity was more consistently increased during stimulation (*Figure 4E*, bottom; *Figure 4—figure supplement 1D*; p=5.88 × 10⁻⁵). These results suggested poor correlation between hyperdirect M1 activity and the behavioral benefits of STN DBS. To further probe the correspondence between DBS effectiveness and hyperdirect M1 activation, we asked if the movement velocity of a single mouse during 60 Hz STN DBS could be predicted by that mouse's M1 calcium activity. We found that movement velocity during DBS did not correlate with change in hyperdirect M1 neural activity (*Figure 4—figure supplement 1F*; $R^2$=−0.14, p=0.96). These findings suggest that while certain stimulation parameters may promote hyperdirect pathway activity, these changes do not strongly correlate with behavioral improvements during DBS.

Additionally, we noted that increases in hyperdirect M1 activity evolved more slowly during STN DBS than increases in STN and SNr activity. The activity of neurons mediating the therapeutic effects of STN DBS would be predicted to change on a similar timescale to behavior. To compare the activation kinetics of STN, SNr, and hyperdirect M1 neurons during DBS, we measured the rise time of the calcium signal in regions and conditions in which we observed significant changes in neural activity: STN (60 and 100 Hz), SNr (60 and 100 Hz), and hyperdirect M1 (100 Hz). Given the observed lag between electrophysiology and bulk GCaMP signals seen by other groups (*Markowitz et al., 2018*), and in our own data (*Figure 1*), we expected that changes in calcium signals might appear to lag the behavior itself. Across all STN DBS conditions, movement velocity increased with a rise time of 2.8±0.6 s after stimulation commenced. For each condition, we calculated the difference in rise time between the calcium signal and movement velocity as an indicator of whether these two signals changed on a similar timescale. STN calcium signals during 60 or 100 Hz STN DBS lagged movement velocity by 2.6±1.1 s. We observed a relatively similar lag comparing SNr calcium signals to the corresponding movement velocity traces (3.7±1.8 s). However, the lag in hyperdirect M1 activity was markedly longer (17.3±3.1 s). These kinetics indicate STN and SNr activity evolve on a timescale similar to the movement benefits of STN DBS, while hyperdirect M1 activity, as measured by fiber photometry, evolves more slowly.

## Surgical removal of M1 does not abolish therapeutic benefit of STN DBS

Though overall hyperdirect pathway activity was not a strong predictor of the therapeutic effects of STN DBS, these findings do not exclude the possibility that the hyperdirect pathway mediates motor benefits. We next asked if M1 was required for the therapeutic effects of STN DBS on movement. We surgically removed the ipsilesional M1 of parkinsonian mice and implanted STN DBS devices (*Figure 5A, Figure 4—figure supplement 1G*). As in previous motor cortex lesion studies (*Kawai*

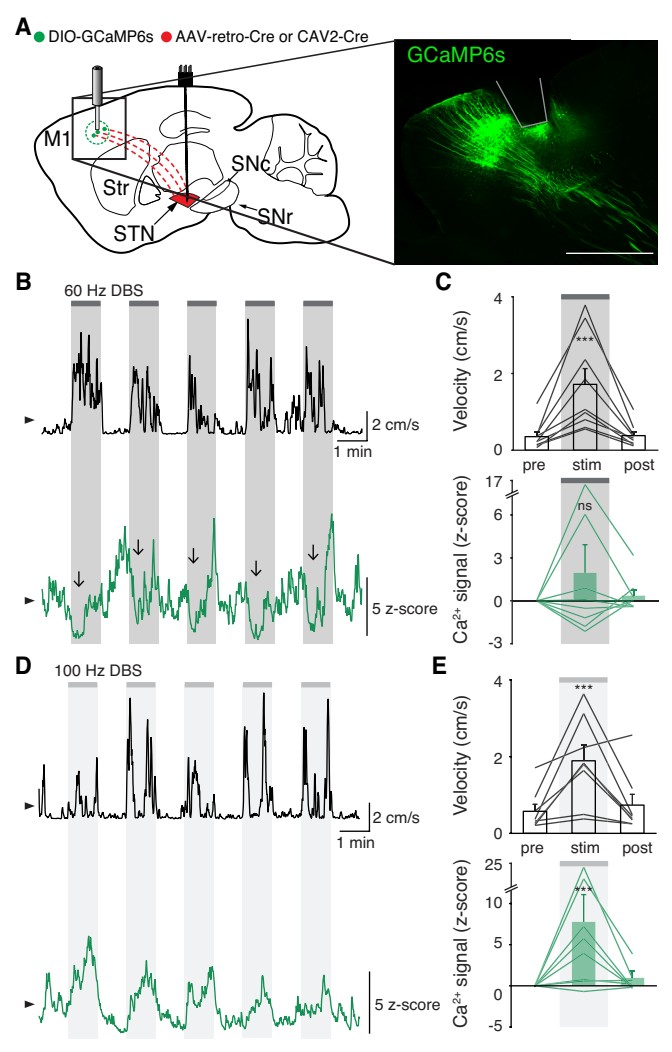

**Figure 4.** Subthalamic nucleus deep brain stimulation (STN DBS) variably changes hyperdirect M1 neural activity. (**A**) Left: Sagittal schematic showing STN DBS and M1-STN GCaMP fiber photometry. Right: Postmortem sagittal section showing GCaMP expression and estimated fiber placement in M1 (inset, scale = 500 μm). (**B**) Representative single-session velocity (black) and M1-STN GCaMP signal (green) in response to 60 Hz STN DBS. (**C**) Average velocity (top) and M1-STN GCaMP signal (bottom) before, during, and after 60 Hz STN DBS (N=9 mice). (**D**) Representative single-session velocity (black) and M1-STN GCaMP signal (green) in response to 100 Hz STN DBS. (**E**) Average velocity (top) and M1-STN GCaMP signal (bottom) before, during, and after 100 Hz STN DBS (N=8 mice). Statistical significance was determined using a one-way repeated measures ANOVA with a Tukey HSD post hoc analysis applied to correct for multiple comparisons; ***p < 0.001 (only comparison between pre and stim shown, see **Supplementary file 1**, table 1 for detailed statistics). Arrowhead in velocity and GCaMP traces corresponds to 1 cm/s and 0 z-score, respectively. Bar plots show mean ± SEM.

The online version of this article includes the following figure supplement(s) for figure 4:

**Figure supplement 1.** STN DBS drives variable and slow changes in hyperdirect M1 activity.

---

*et al., 2015*), mice were allowed to recover for at least 10 days before behavioral testing. Remarkably, these mice still showed a significant increase in movement velocity in response to 100 Hz STN DBS (*Figure 5B–C*; p=8.24 × 10⁻³). Thus, it is unlikely that antidromic activation of M1 is the primary driver of the therapeutic benefit of STN DBS in parkinsonian mice.

## STN movement-related activity is disrupted by therapeutic STN DBS

Our results indicate that STN DBS is unlikely to work through inhibition of basal ganglia output, nor solely through antidromic excitation of M1. However, a third possibility is that STN DBS disrupts the

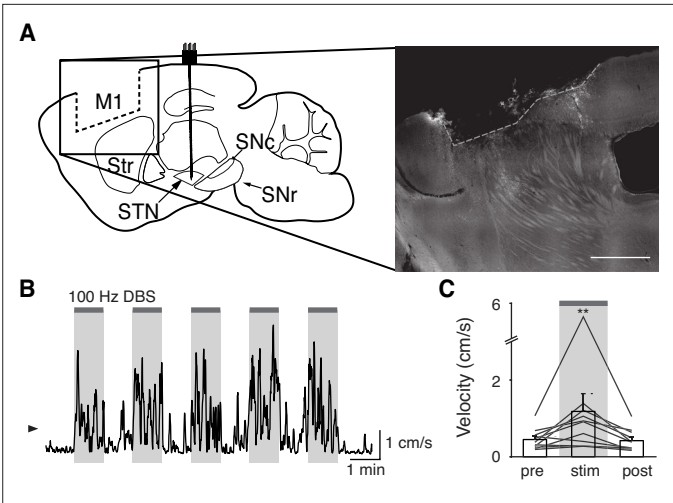

**Figure 5.** Surgical removal of M1 does not abolish therapeutic benefit of subthalamic nucleus deep brain stimulation (STN DBS). (**A**) Left: Sagittal schematic showing STN DBS and M1 surgical lesion. Right: Postmortem sagittal section showing estimated extent of M1 lesion (inset, scale = 500 μm). (**B**) Representative single-session velocity of an M1-lesioned hemiparkinsonian mouse in response to 100 Hz STN DBS. (**C**) Average velocity before, during, and after 100 Hz STN DBS in M1-lesioned hemiparkinsonian mice (N=11 mice). Statistical significance was determined using a one-way repeated measures ANOVA with a Tukey HSD post hoc analysis applied to correct for multiple comparisons; **p < 0.01 (only comparison between pre and stim shown, see ***Supplementary file 1***, table 1 for detailed statistics). Arrowhead in velocity trace corresponds to 1 cm/s. Bar plots show mean ± SEM.

pattern of neural activity within the STN itself. As previously noted, in parkinsonian mice, STN activity increases around movement initiation (***Figure 6A***, left), consistent with the idea that STN neurons encode some aspects of movement. To determine whether this encoding was altered during DBS, we aligned neural activity to movement starts during stimulation epochs. We found that although the overall average calcium signal was increased during DBS (***Figure 2***), the movement-aligned increase in STN activity was abolished during therapeutic STN DBS at 100 Hz (***Figure 6A***, right). This observation suggested that STN movement encoding was disrupted by STN DBS.

This disruption might result from any patterned electrical stimulation, or it could represent a direct correlate of therapeutic stimulation. To test these possibilities, we chose 13 parameter combinations (varying in current amplitude, frequency, and pulse width, ***Supplementary file 1***, table 3) from a set that we had used previously to evaluate the impact of STN DBS parameters on behavioral benefit in the mouse model (***Schor and Nelson, 2019***). In the same mice, we measured locomotor activity and calcium signals while delivering STN DBS at each of these 13 parameter combinations. Behavioral responses were divided into two groups, depending on whether the DBS-induced movement velocity averaged greater or less than 1 cm/s (***Figure 6B***). We labeled the first group 'high effect' stimulation (8 combinations) and the second group 'low effect' stimulation (5 combinations). We then assessed STN movement-related activity during each stimulation type, as measured by the peak-to-trough deflection of the movement-aligned STN photometry signal (***Figure 6C***, shaded inset). Interestingly, across 13 stimulation parameter sets, this neural activity metric showed a bimodal distribution. With low effect stimulation parameters, STN calcium signals increased around movement onset, as they did during baseline (no stimulation) periods. This resulted in a peak-to-trough change in calcium that was >1 (z-scored dF/F; ***Figure 6C***). However, during high effect stimulation, STN calcium signals changed minimally around movement onset, with a peak-to-trough change of <1 (***Figure 6C***). When comparing STN activity across all high effect vs. all low effect stimulation parameters rather than individually, the normal movement-related increase in STN activity was strongly suppressed only during highly effective stimulation (***Figure 6D***; p=1.48 × 10$^{-11}$ for pre vs. high, p=0.54 for pre vs. low). Taken together, these results suggest that STN DBS disrupts movement-related STN activity and furthermore that this disruption is specific to behaviorally beneficial stimulation parameters.

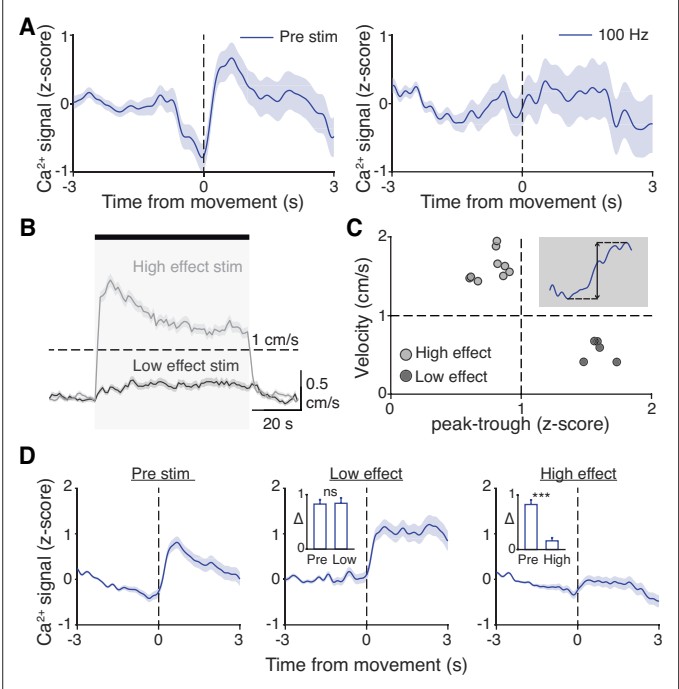

**Figure 6.** Subthalamic nucleus (STN) movement-related activity is disrupted by therapeutic STN deep brain stimulation (DBS). (**A**) Average STN fiber photometry signal aligned to movement starts during pre-stim periods (left) and during 100 Hz STN DBS stimulation (right). (**B**) Average movement velocity over time during high effect stimulation (light gray, achieves average velocity >1 cm/s) and low effect stimulation (dark gray, achieves average velocity <1 cm/s). (**C**) Scatter plot comparing peak-trough distance in the average movement-aligned photometry trace (inset) to velocity during high effect (light gray) and low effect (dark gray) stimulation parameters. Each dot represents an average across all mice for each stimulation parameter (N=8 mice). (**D**) Average STN fiber photometry signal aligned to movement starts during pre-stim periods (left) and during low effect STN DBS stimulation (middle) and high effect STN DBS stimulation (right). Inset bar graphs show difference between average fiber signal in 1 s following movement start and 1 s preceding movement start (see Materials and methods for further details) during pre, low effect, or high effect stim. Statistical significance was determined using a Wilcoxon rank-sum test; \*\*\*p < 0.001 (see *Supplementary file 1*, table 1 for detailed statistics). Bar plots, photometry, and velocity traces show mean ± SEM.

## Disruption of movement-related STN activity is sufficient to provide therapeutic benefit

STN DBS may trigger many changes in both the rate and timing of neural activity. However, it remains critical to determine which changes in neural activity causally contribute to the therapeutic mechanism(s) of STN DBS. In our experiments using electrical STN DBS, we observed changes in overall STN activity <u>and</u> activity around movement initiation, making it difficult to determine which change is more likely to mediate the benefit. To disentangle the behavioral impacts of changing STN rate vs. timing, we replaced electrical DBS/optical recording with optical DBS/electrical recording techniques. We injected parkinsonian VGlut2-Cre mice with Cre-dependent channelrhodopsin (ChR2) and implanted 16-channel optrode arrays in the STN (*Figure 7A*, *Figure 7—figure supplement 1A*). We then assessed STN single-unit firing during two optical stimulation paradigms in freely moving parkinsonian mice: 'constant' and 'pulsatile' (50 Hz) blue light stimulation (both 3 mW). As has been observed previously in anesthetized STN recordings in rats (*Yu et al., 2020*), STN neurons showed both excitatory and inhibitory responses to optical stimulation (*Figure 7B–E*; *Figure 7—figure supplement 1B-C*). However, both stimulation paradigms caused similar decreases in overall firing rate (*Figure 7C and E*; p=0.98 comparing relative firing rates during constant and 50 Hz stimulation, Wilcoxon signed-rank test). Of note, the response of STN neurons to optical stimulation differed between in vivo/freely moving and ex vivo preparations. During cell-attached patch-clamp recordings of STN in ex vivo slices (*Figure 7—figure supplement 1D*), pulsatile (50 Hz) blue light stimulation increased STN firing

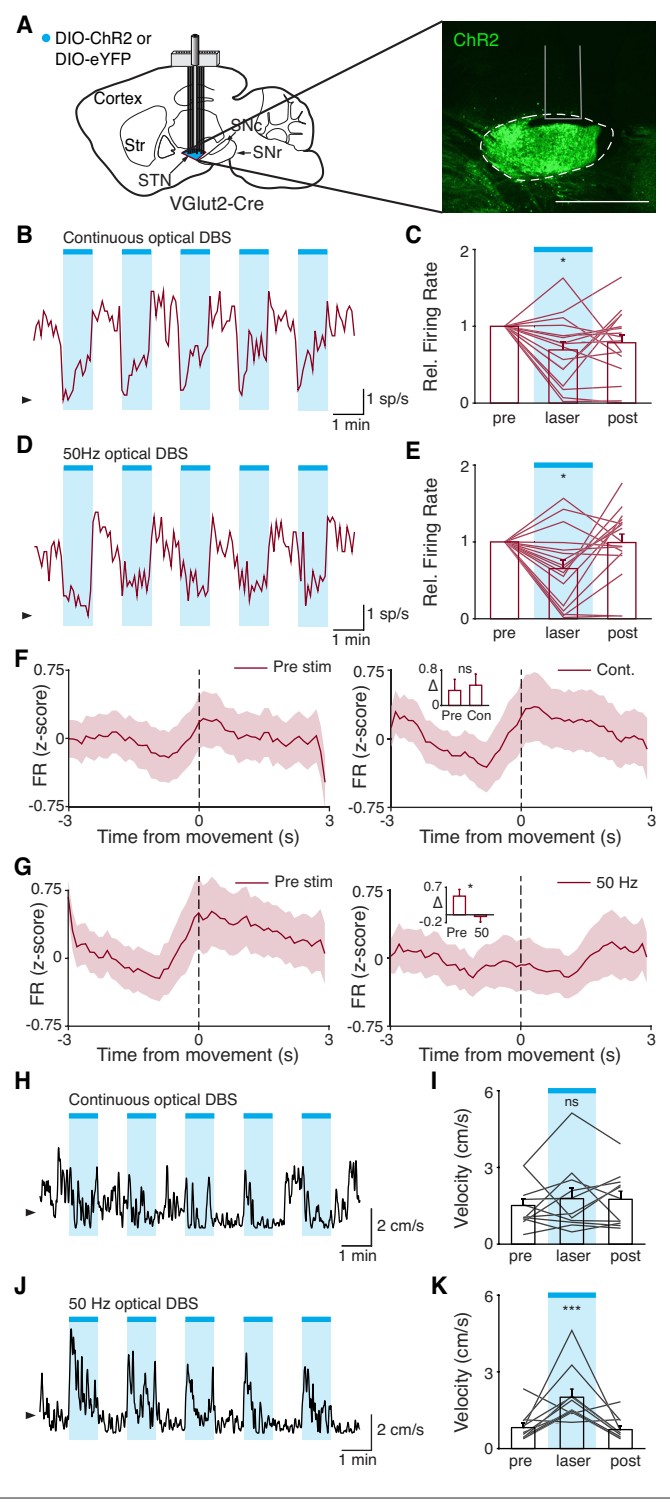

**Figure 7.** Disruption of movement-related subthalamic nucleus (STN) activity is sufficient to provide therapeutic benefit. (**A**) Left: Sagittal schematic showing viral injection and 16-channel optrode implantation in the STN. Right: Postmortem sagittal section showing ChR2 expression in the STN (inset, scale = 500 μm). (**B**) Representative STN single-unit firing in response to continuous optical stimulation, given in 1 min epochs (blue bars). (**C**) Average z-scored firing rate of STN single units before, during, and after continuous optical stimulation (n=17 neurons, N=3 mice). (**D**) Representative STN single-unit firing in response to 50 Hz optical stimulation, given in 1 min epochs (blue bars). (**E**) Average z-scored firing rate of STN single units before, during, and after 50 Hz optical stimulation

*Figure 7 continued on next page*

*Figure 7 continued*

(n=17 neurons, N=3 mice). (**F**) Average z-scored firing rate of STN single units aligned to movement starts before (left) and during (right) continuous optical stimulation. Inset: average change in firing rate around movement starts (see Materials and methods for further details) before (pre) or during (con) constant optical stimulation (n=17 neurons, N=3 mice). (**G**) Average z-scored firing rate of STN single units aligned to movement starts before (left) and during (right) 50 Hz optical stimulation. Inset: average change in firing rate around movement starts before (pre) and during (50) 50 Hz optical stim (n=17 neurons, N=3 mice). (**H**) Representative single-session movement velocity in response to continuous optical stimulation, given in 1 min epochs. (**I**) Average velocity before, during, and after continuous optical stimulation (N=11 mice). (**J**) Representative single-session movement velocity in response to 50 Hz optical stimulation, given in 1 min epochs. (**K**) Average velocity before, during, and after 50 Hz optical stimulation (N=11 mice). Statistical significance was determined using a Wilcoxon rank-sum test (**F–G**) or a one-way repeated measures ANOVA with a Tukey HSD post hoc analysis applied to correct for multiple comparisons (**C,E,I,K**); *p<.05, ***p < 0.001 (for ANOVA, only comparison between pre and laser shown, see ***Supplementary file 1***, table 1 for detailed statistics). Arrowhead in firing rate and velocity traces corresponds to 0 spike/s and 1 cm/s, respectively. Bar plots and average firing rate traces show mean ± SEM.

The online version of this article includes the following figure supplement(s) for figure 7:

**Figure supplement 1.** In vivo and ex vivo optical STN stimulation produces distinct changes in firing rate.

---

fairly consistently (***Figure 7—figure supplement 1E-F***; p=0.033), while constant illumination evoked more variable changes in spiking (***Figure 7—figure supplement 1G-H***; p=0.63). The effects of optical stimulation on STN firing in the in vivo, freely moving condition, however, are most relevant to the behavioral impact of STN DBS.

While both optical stimulation patterns produced decreases in the average firing rate of STN neurons, the two stimulation paradigms had different effects on movement-related STN activity. During stimulation-off epochs for both paradigms, STN firing rate increased around the time of movement starts (***Figure 7F and G***, left), as we had observed previously (***Figure 1L***). During constant optical stimulation, these movement-related firing dynamics did not change significantly (***Figure 7F***, right; p=0.51). In contrast, pulsatile 50 Hz stimulation greatly attenuated firing rate increases around movement onset (***Figure 7G***, right; p=0.039). Based on these electrophysiological recordings, continuous and 50 Hz optical stimulation produce similar changes in firing rate, but distinct effects on movement-related activity, allowing us to disentangle the effects of STN rate and pattern in producing therapeutic effects in parkinsonian mice.

To test whether these two stimulation paradigms would produce different behavioral effects, we studied a larger cohort of parkinsonian VGlut2-Cre mice, injected with Cre-dependent ChR2 or eYFP (control) and implanted with optical fibers in the STN (***Figure 7—figure supplement 1A***). In different test sessions, we stimulated with either constant or pulsatile blue light in 1 min epochs. We predicted that continuous stimulation, which did not alter the trajectory of STN firing around movement initiation, would not produce benefits, while pulsatile stimulation, which greatly attenuated STN these dynamics, would increase movement velocity. During constant illumination, neither STN-ChR2 nor STN-eYFP mice showed a significant increase in movement velocity or other metrics such as percent time moving or change in rotation bias (***Figure 7H–I***; ***Figure 7—figure supplement 1I-K***; p=0.30 for ChR2 movement velocity, p=0.66 for eYFP movement velocity). However, STN-ChR2 mice receiving 50 Hz stimulation did show a significant increase in movement velocity and percent time moving (***Figure 7J–K***; ***Figure 7—figure supplement 1L-M***; p=1.27 × 10$^{-8}$ for movement velocity, p=5.56 × 10$^{-9}$ for percent time moving), similar to mice receiving electrical STN DBS, and also developed an increased bias toward ipsilesional rotations (***Figure 7—figure supplement 1L*** (inset); p=0.016). STN-eYFP mice did not show a significant change in movement velocity during 50 Hz stimulation (***Figure 7—figure supplement 1N***; p=0.81). These results demonstrate that although therapeutic electrical and optical STN DBS produce quite distinct changes on the rate of STN activity, they both disrupt movement-related STN dynamics. Moreover, they suggest that disrupting movement-related STN dynamics is sufficient to ameliorate parkinsonian motor deficits, making it a candidate mechanism for electrical STN DBS.

## Discussion

We combined a recently developed mouse model of electrical STN DBS for PD with electrical artifact-free GCaMP fiber photometry to investigate three major theories surrounding the mechanism of electrical STN DBS: (1) STN DBS inhibits STN and SNr activity; (2) STN DBS acts through antidromic activation of hyperdirect M1 neurons; and (3) STN DBS acts through disrupting neuronal activity patterns within the STN. Using bulk calcium signals as a surrogate marker for neural activity, we found that STN DBS increases rather than decreases STN and SNr activity, and M1 activation does not seem to be necessary for therapeutic STN DBS, rendering the first two theories less likely. In support of the third theory, we observed that therapeutic electrical stimulation abolished patterned STN activity around movement onset and used optical manipulations to demonstrate that attenuating STN dynamics may be causal in the motor benefits of DBS.

In this study, we used fiber photometry with GCaMP6s to examine how neural activity changes during STN DBS. A recent study demonstrated technical feasibility of using in vivo calcium imaging with electrical STN DBS (*Trevathan et al., 2021*), but ours is the first to use this approach to link DBS-mediated physiological changes to behavior in freely moving parkinsonian animals. Moreover, our study is one of the first to use fiber photometry in deep basal ganglia nuclei, and to compare these signals with in vivo single-unit electrophysiology. This approach had several advantages, as well as limitations. The key advantage was the ability to obtain recordings free from electrical artifacts. Artifacts had been a major obstacle in prior electrophysiological studies, particularly in studying the effect of DBS on neurons in the target structure, such as the STN or GPi. A second advantage was the ability to use cell type- or projection-specific techniques. For example, the use of retrograde viruses allowed us to target the direct projections from primary motor cortex to STN (hyperdirect pathway), a population of significant interest in PD and DBS. Though STN and SNr are relatively homogeneous structures, with regard to major neurotransmitters (*Smith and Parent, 1988*; *Walaas and Fonnum, 1979*), future studies could use GCaMP and genetics to target either specific STN/SNr projections or novel cell types within them (*Kita and Kitai, 1987*; *Liu et al., 2020*). There are several limitations of our approach over electrophysiology, however. One is its temporal resolution. While traditional single-unit electrophysiology can detect individual action potentials, measuring calcium with GCaMP provides an integrated signal arising from multiple spikes (*Chen et al., 2013*; *Sabatini, 2019*). Fiber photometry further averages signals across a population of neurons, and from multiple neuronal compartments (dendrites and somata), bringing additional caveats. Fiber photometry signals may increase in response to greater firing in the population as a whole, or in response to increased synchronization across the population. Recent work comparing neuronal firing and fiber photometry (with simultaneous multi-unit electrophysiology and fiber photometry) in the striatum suggests that while an initial phase of calcium signal correlates well with firing, fiber photometry signals are also more prolonged, which may represent dendritic calcium influx associated with backpropagation (*Legaria et al., 2021*). For a combination of reasons, fiber photometry signals are slow and often lag changes in spiking. Given this temporal lag, it is difficult to establish whether DBS-associated changes in GCaMP signal arise directly from electrical stimulation, network activation, sensory feedback, or all of the above. Nonetheless, our activity measurements with photometry showed striking similarities to activity measurements using single-unit electrophysiology, over both short (movement-aligned activity) and long (responses to systemic levodopa administration) timescales. There thus appears to be some correspondence between the two recording techniques. In the future, voltage indicators with high signal-to-noise that are compatible with deep imaging, as well as miniscope imaging of many single neurons simultaneously, may allow detection of single spikes and help increase the information obtained in optical recordings during DBS.

Our observation that therapeutic STN DBS increases activity at the level of the STN and SNr, rather than inhibiting it, is at odds with traditional rate-based models of basal ganglia function. In fact, previous STN DBS electrophysiological studies in parkinsonian primates have had conflicting results: some show increased basal ganglia output activity during STN DBS, while others observe local STN inhibition (*Filali et al., 2004*; *Moran et al., 2011*). Single-unit data in rodents is more limited, but analyses of pattern at both the single-unit and LFP level in one study suggest that the therapeutic benefit of STN DBS may be independent from overall changes in STN firing rate (*Zhuang et al., 2018*). Thus, it is possible that treatments that increase (e.g. DBS) or decrease STN firing (e.g. optical inhibition

[*Yoon et al., 2014*] or levodopa) may be therapeutic. Some discrepancies in existing physiological data may arise from the imperfect process of removing STN DBS artifacts from electrophysiological recordings, especially in structures like STN and SNr that have high spontaneous firing rates. Others may relate to differences among animal models of PD.

Electrical STN DBS may increase the activity of STN neurons via several physiological mechanisms. As electrical stimulation has previously been suggested to preferentially recruit axons (*McIntyre et al., 2004*; *McIntyre and Grill, 1999*; *Nowak and Bullier, 1998*), DBS may drive antidromic spiking of STN neurons via stimulation of efferent STN axons, as well as drive STN spiking via activation of incoming excitatory axons. Depending on the proximity of the electrode to neural elements, it may drive STN activity through local activation of dendrites or cell bodies, as well. Increased SNr spiking may be driven by orthodromic activation of STN to SNr excitatory axons, or potentially by more complex network effects. The time resolution of fiber photometry and the time lag between STN and SNr activation (on average about 1 s) with this method cannot exclude polysynaptic effects. It is important to note that fluorescence GCaMP signals are correlative, and the behavior of mice during therapeutic STN DBS changes markedly. The change in movement may itself drive changes in STN and SNr activity, for example through sensory feedback to the basal ganglia. The recorded signals in the STN and SNr may therefore be both a cause and/or an effect of increased movement.

Though we observed modulation of hyperdirect M1 neurons during STN DBS, this modulation did not correlate well with therapeutic effects. While this is suggestive that the hyperdirect pathway is less important for the therapeutic effect of STN DBS, differences in the physiological properties of M1 projection neurons, as compared to STN and SNr neurons, might have contributed to the late-developing changes in M1. The spatial relationship of the DBS electrodes to the M1-STN projection, and the targeting of GCaMP to the hyperdirect pathway are likely to vary across animals, which may have also contributed to variability in hyperdirect pathway signals during DBS. In addition, antidromic activation of hyperdirect pathway neurons may have more complex polysynaptic effects, via local interneurons or collaterals to different areas, such as the zona incerta (*Kita and Kita, 2012*). However, removal of M1 did not abolish the therapeutic benefit of STN DBS, leading us to conclude that hyperdirect M1 activation may not play a central role. The discrepancy between our conclusions and those of previous studies has a number of potential explanations. While past mouse studies have used optogenetic stimulation as a proxy for electrical STN DBS (*Gradinaru et al., 2009*; *Sanders and Jaeger, 2016*), we used electrical stimulation in an effort to more closely model what is observed in PD patients (our eventual use of optogenetic stimulation was directly informed by our observations during electrical stimulation). The former approach identifies manipulations that are sufficient to relieve parkinsonian motor symptoms, while the latter identifies changes that correlate with a specific therapy. Thus, while optogenetic stimulation may reveal that changing neural activity in a variety of ways can relieve parkinsonism in mice, it is difficult to extrapolate which of these changes actually occur during electrical STN DBS. In other words, two therapies that have similar behavioral effects may not have the same mechanism of action. In addition, while hyperdirect pathway neurons did not seem to be crucial for the benefits of STN DBS in mice, they may play a more important role in other species, particularly given differences in the connectivity of motor cortical areas and STN in primate species (*Devergnas and Wichmann, 2011*; *Miocinovic et al., 2018*). Indeed, it is important to note that animal models of PD, such as the 6-OHDA model, are different from PD, and caution is required in extrapolating our results to DBS in PD patients. Additionally, STN DBS relieves both gross and fine motor symptoms in PD, but here we have only assessed gross motor outcomes. It is possible that hyperdirect pathway modulation contributes more strongly to fine motor benefits of STN DBS.

In the absence of therapeutic manipulations, we found that STN neurons in parkinsonian mice show an increase in activity around movement initiation, as measured with single-unit electrophysiology or bulk calcium imaging with GCaMP fiber photometry. Using electrophysiological methods, similar observations have been made in the STN of healthy NHPs and cats (*Cheruel et al., 1996*; *Wichmann et al., 1994*). Less is known about what drives this increase, though candidates include excitatory input from hyperdirect M1 neurons (*Polyakova et al., 2020*), inhibitory input from the globus pallidus pars externa (*Chu et al., 2015*; *Kovaleski et al., 2020*), and/or increased synchronization among STN neurons. As we found that disruption of STN movement-related activity may play a role in alleviating

parkinsonian symptoms, identifying the physiological sources of this signal will likely be an important task in future research.

Our observation that movement-related STN activity patterns are altered during STN DBS may relate to previous work showing changes in firing patterns or network synchrony during STN DBS. It has previously been postulated that rate-independent aspects of neural activity, such as within-neuron firing pattern or between-neuron synchronization, may drive PD symptoms and represent key markers of therapeutic interventions (*Hammond et al., 2007*; *Little and Brown, 2014*; *Wichmann, 2019*). Many other groups have observed increased oscillations throughout the basal ganglia in parkinsonian animal models and in humans, which may resolve with therapeutic treatment (*de Hemptinne et al., 2015*; *Halje et al., 2012*; *Moran et al., 2012*; *Shimamoto et al., 2013*). In fact, one group studying healthy NHPs has even observed positive modulation during movement in pallidal cells, similar to what we observed during movement starts in STN neurons, and which was similarly interrupted in a subset of pallidal neurons during STN DBS (*Zimnik et al., 2015*). The difficulty, though, has been in establishing a causal link between changes in these patterns during DBS and improvement in behavior. Our ability to not only observe but also to trigger these changes through optogenetic manipulation helps fill this critical gap. Additional causal links might be investigated further in the future using a combination of optical and electrical methods, building on the approach introduced here.

Though it facilitated our use of cell type-specific methods and GCaMP, a key potential caveat of our study is the mouse model of PD. First, the 6-OHDA model causes focal, rapid, and in this case nearly complete loss of dopaminergic neurons and their projections. In contrast, PD causes neurodegenerative changes in multiple brain areas over many years. Though many key physiological features of PD are similar in toxin-based models of parkinsonism (*Bové and Perier, 2012*; *Campos et al., 2013*), others may be distinct based on the pattern and tempo of neurodegeneration. In addition, electrical stimulation may have different impacts in the small mouse brain as compared to the human brain. Though electrical stimulation is unlikely to respect the borders of brain nuclei in either species, the small size of the target region (STN) in the mouse brain increases the likelihood that fibers in adjacent areas are recruited by DBS. Comfortingly, our observation that the therapeutic effects of electrical STN DBS in the mouse fall off rapidly below the average current amplitude (200 μA) and vary according to the STN region targeted (*Schor and Nelson, 2019*) match well with human data and suggest some specificity in the volume of tissue activated (*Greenhouse et al., 2011*; *Tommasi et al., 2008*).

For practical reasons, in order to test the physiological and behavioral effects of a wide variety of stimulation parameters (both subtherapeutic and therapeutic), we delivered short (1 min) epochs of stimulation. These short epochs may not capture the longer term changes in neural activity expected in PD patients, where continuous high-frequency stimulation is the current clinical standard. Reducing this concern, many DBS benefits indeed evolve rapidly in PD patients, as in the mouse model (*Hristova et al., 2000*; *Temperli et al., 2003*), and our previous work demonstrates consistent behavioral improvement in the model even across longer timescales (*Schor and Nelson, 2019*).

Excitingly, our observation that non-canonical changes in STN activity confer therapeutic benefit in a mouse model of PD suggests a wider therapeutic space for the treatment of PD. Many therapeutic approaches to PD have been predicated on the idea that inhibition of hyperactive basal ganglia nuclei is required for therapeutic benefit, but our findings, as well as recent work using closed-loop DBS (*Bouthour et al., 2019*; *Johnson et al., 2016*; *Rosin et al., 2011*) indicate non-rate-based alterations in activity can improve movement. In addition, our work linking neural activity to behavior in STN DBS for PD may inform the application of DBS to other neuropsychiatric disorders. To rationally apply DBS to other conditions, such as addiction or Tourette's syndrome, it is critical to know how electrical stimulation might impact the underlying neural circuitry of disease. We hope that our work will promote future inquiries into the therapeutic potential of DBS.

# Materials and methods

## Key resources table

| Reagent type (species) or resource | Designation | Source or reference | Identifiers | Additional information |
|---|---|---|---|---|
| Genetic reagent (*Mus musculus*, C57Bl/6) | VGluT2-Cre | JAX | RRID IMSR_JAX:028863 | Hemizygous |
| Genetic reagent (*Mus musculus*, C57Bl/6) | VGAT-Cre | JAX | RRID IMSR_JAX:016962 | Hemizygous |
| Strain, strain background (*Mus musculus*, C57Bl/6) | C57Bl/6J | JAX | RRID IMSR_JAX:000664 | Wild-type |
| Genetic reagent (AAV) | AAV1-Syn-Flex-GCaMP6s-WPRE-SV40 | Addgene | RRID:Addgene_100845 | Diluted 1:8 in normal saline |
| Genetic reagent (AAV) | AAV2$_{retro}$-Cre-mCherry | Addgene | RRID:Addgene_55632 | Undiluted |
| Genetic reagent (CAV) | CAV-Cre | Montpellier | CAV-Cre | Undiluted |
| Genetic reagent (AAV) | AAV5-DIO-ChR2-eYFP | Addgene | RRID:Addgene_20298 | Diluted 1:2 in normal saline |
| Genetic reagent (AAV) | AAV5-DIO-eYFP | UNC Vector Core | No. order # available | Undiluted |
| Other | 16-Channel 7 mm fixed electrode array | Innovative Neurophysiology | No. catalog # | Tungsten microwire array |
| Other | Master8 | AMPI | RRID:SCR_018889 | Stimulus pattern generator |
| Antibody | Anti-tyrosine hydroxylase (rabbit polyclonal) | Pel-Freez | Order # P40101-150 RRID:AB_2617184 | lot #: AJ0818B; (1:1000) dilution |
| Antibody | Anti-tyrosine hydroxylase (chicken polyclonal) | Millipore | Order # AB9702 RRID:AB_570923 | lot #: 3152195; (1:1000) dilution |
| Software, algorithm | Plexon Offline Sorter | Plexon | RRID:SCR_000012 | |
| Software, algorithm | Ethovision | Noldus | RRID:SCR_000441 | |
| Software, algorithm | IgorPro | Wavemetrics | RRID:SCR_000325 | With MafPC: https://www.xufriedman.org/mafpc |
| Software, algorithm | Matlab | Mathworks | RRID:SCR_001622 | v.2019a |

## Animals

Three- to six-month-old WT and transgenic C57Bl/6 mice of either sex were used in this study. To allow optical recording and manipulation of glutamatergic STN neurons, homozygous VGlut2-Cre mice (Stock No. 028863; RRID IMSR_JAX:028863, Jackson Labs) were bred to WT C57BL/6 mice (Jackson Labs) to yield hemizygous VGlut2-Cre mice. To allow optical recording of GABAergic SNr neurons, homozygous VGAT-Cre mice (Stock No. 028862; IMSR_JAX:016962, Jackson Labs) were bred to WT C57BL/6 mice (Stock No. 000664; RRID IMSR_JAX:000664, Jackson Labs) to yield hemizygous VGAT-Cre mice. Hemizygous experimental animals were used to minimize potential confounds reported in association with homozygous BAC transgenic animals or homozygous breeding schemes (*Chan et al., 2012*; *Nelson et al., 2012*). VGluT2-Cre and VGAT-Cre mice have IRES-Cre targeted to the *Slc17a6* locus and *Slc32a1* locus, respectively. Animals were housed 1–5 per cage on a 12 hr light/dark cycle with ad libitum access to rodent chow and water. All behavioral manipulations were performed during the light phase. We complied with local and national ethical regulations regarding the use of mice in research. All experimental protocols were approved by the UC San Francisco Institutional Animal Care and Use Committee.

## Electrical DBS devices

A detailed protocol for DBS implants can be found at https://doi.org/10.17504/protocols.io.261gen92dg47/v1. Briefly, we constructed electrical DBS devices consisting of three twisted pairs of stainless steel wire (76.2 μm diameter, coated, AM Systems), cut at an angle to span approximately 300 μm in DV. These were pressure-fit into female Millmax connectors. Each electrode pair was tested for shorts prior to electrode implantation. For additional details, see *Schor and Nelson, 2019*.

## Surgical procedures

A detailed protocol for stereotaxic surgery can be found at https://doi.org/10.17504/protocols.io.n2bvj6qynlk5/v1. Briefly, stereotaxic surgery was performed between 3 and 6 months of age. Anesthesia was induced with intraperitoneal (IP) injection (0.1 mL) of ketamine (40 mg/kg) and xylazine (10 mg/kg) and maintained with inhaled isoflurane (0.5–1%). To model PD in mice, the neurotoxin 6-OHDA (1 μL, 5 mg/mL) was injected unilaterally in the left MFB (–1.0 AP, –1.0 ML, 4.9 DV from Bregma). Desipramine (0.2 mL, 2.5 mg/mL) was injected IP approximately 30 min prior to 6-OHDA injections to reduce uptake by other monoaminergic neurons in the MFB. Additional surgeries were performed at least 2 weeks following 6-OHDA injection.

For experiments involving combined electrical STN DBS and optical imaging, a three-lead bipolar stimulating electrode array was implanted in the ipsilesional STN (–1.8 AP, –1.65 ML, 4.5 DV) (*Schor and Nelson, 2019*). During the same surgery, VGlut2-Cre mice were (RRID: Addgene_100845; 100 nL injected diluted 1:8 in normal saline, undiluted titer $3.06 \times 10^{13}$ /mL) in the STN (–1.8 AP, –1.65 ML, 4.5 DV) and implanted with a photometry fiber-optic ferrule (0.4 mm, Doric Lenses) above the STN (4.3 DV). VGAT-Cre mice were injected with the same Cre-dependent GCaMP6s vector (300–500 nL injected diluted 1:8 in normal saline) in the SNr (–3.2 AP, –1.6 ML, 4.5 DV) and implanted with a fiber-optic ferrule above the SNr (4.3 DV). WT mice were injected with a retrograde virus encoding Cre recombinase (either CAV-Cre [Montpellier, 100 nL injected undiluted, undiluted titer $1.0 \times 10^{13}$ /mL; https://plateau-igmm.pvm.cnrs.fr/?vector=cav-cre] or AAV2$_{retro}$-Cre-mCherry [Addgene/UPenn Vector Core RRID Addgene_55632, 100 nL injected undiluted, undiluted titer $7.8 \times 10^{13}$ /mL]) in the STN (–1.8 AP, –1.65 ML, 4.5 DV) and Cre-dependent GCaMP6s (500 nL injected diluted 1:8 in normal saline) in the primary motor cortex (M1, +2 AP, –1.56 ML, 1 DV) and implanted with a fiber-optic ferrule above M1 (0.8 DV).

For in vivo electrophysiological experiments, mice were implanted with a 16-channel 7 mm fixed electrode array (Innovative Neurophysiology) in either the STN or the SNr, using the same coordinates as used for GCaMP6s injections above. For combined in vivo electrophysiology/optical stimulation experiments, VGlut2-Cre littermates were injected with Cre-dependent AAV5-DIO-ChR2-eYFP (UPenn/Addgene, RRID Addgene_20298, injected diluted 1:2 in normal saline, 100 nL, undiluted titer $1.02 \times 10^{13}$ /mL) or AAV5-DIO-eYFP (UNC, no order # available; injected undiluted, 100 nL, titer $4.4 \times 10^{12}$ /mL) in a randomized fashion, and implanted with a 16-channel 7 mm fixed electrode array (Innovative Neurophysiology) with a fiber-optic ferrule (0.2 mm, Thor Labs) epoxied ~0.2 mm above the electrode tips in the STN. For optical stimulation experiments without recording, a fiber-optic ferrule was implanted just above the STN (4.3 DV). A minimum of 3 weeks were allowed for viral expression before behavioral testing.

For experiments involving M1 lesioning, a large rectangular craniectomy was performed to expose brain tissue containing M1 (vertices of rectangle at [–0.1 AP, –2.1 ML]; [2.6 AP, –2.1 ML]; [2.6 AP, –0.9 ML]; and [–0.1 AP, –0.9 ML]). A micro knife (FST) was then used to carefully remove a 1 mm thick rectangle of brain tissue that was the height and width of the craniectomy under a dissecting microscope. A hemostatic sponge (Ethicon) was used to staunch any bleeding before covering the lesion and adjacent bone with silicone sealant (Kwik-Cast). A three-lead bipolar stimulating electrode array was then implanted in the ipsilesional STN as previously described. The scalp was closed with Vicryl suture. The analgesics buprenorphine (0.05 mg/kg IP) and ketoprofen (5 mg/kg SQ) were administered immediately postoperatively and as needed subsequently for postoperative pain. A minimum of 10 days of recovery was allowed before subsequent behavioral testing.

## Behavior

A protocol for behavioral testing can be found here: https://doi.org/10.17504/protocols.io.n2bvj6qynlk5/v1. All behavior was conducted in the open field (clear acrylic cylinders, 25 cm diameter) following 1 day of habituation (20 min). Mice were monitored via two cameras, one directly above and one in front of the chamber. Video-tracking software (Noldus Ethovision XT version 10; RRID:SCR_000441) or custom-written code (Matlab v. 2019a RRID:SCR_001622) was used to quantify locomotor activity, including movement velocity, ipsilateral rotations, and contralateral rotations. Dyskinesia was scored manually by an unblinded rater using a modified version of the abnormal involuntary movement scoring method (*Cenci and Lundblad, 2007*). Dyskinesia was quantified in 1 min increments either every minute (for STN DBS experiments) or every 5 min (for levodopa experiments),

with axial, limb, and orofacial body segments rated on a scale of 0–3 each. A score of 0 indicates no abnormal movement, while a score of 3 indicates continuous dyskinesia for the 1 min epoch. The scores for each body segment are then summed, with a maximum score of 9 per epoch.

## Pharmacology

6-OHDA (Sigma Aldrich) was prepared at 5 mg/mL in normal saline. Levodopa was prepared (0.5 mg/mL Sigma Aldrich) with benserazide (0.25 mg/mL, Sigma Aldrich) in normal saline and always administered at 5 mg/kg.

## Electrical stimulation

An isolated constant current bipolar stimulator (WPI) was used to deliver electrical stimuli. The timing of stimuli was controlled by TTL input from an Arduino. Electrical stimulation experiments consisted of five 1 min stimulation periods, each preceded and followed by 1 min of no stimulation, for a total of 11 min. Both the construction of STN DBS electrodes and the determination of optimal stimulation electrode pair were as detailed previously (*Schor and Nelson, 2019*).

## Fiber photometry

A detailed protocol for fiber photometry can be found at https://doi.org/10.17504/protocols.io.8epv59dbjg1b/v1. Fiber photometry signals were acquired through implanted 400 µm optical fibers, using an LED driver system (Doric). Following signal modulation, 405 nm (control signal, from GCaMP autofluorescence) and 465 nm signals were demodulated via a lock-in amplifier (RZ5P, TDT), visualized, and recorded (Synapse, TDT). Offline, the 405 nm signal was fit to the 465 nm signal using a first-degree polynomial fit (Matlab) to extract the non-calcium-dependent signal (due to autofluorescence, fiber bending, etc.). The fitted 405 nm signal was then subtracted from the 465 nm signal to generate a motion-corrected signal. Animals in which the fitted 405 nm signal did not differ from the 465 nm signal were excluded from further analysis. To remove the gradual, slow bleaching observed in the ~3 hr saline and levodopa recordings, we additionally fit a double exponential to the 405 nm signal, linearly fit it to the the motion-corrected signal, and then subtracted it.

Every processed fiber photometry signal was normalized (z-scored) by subtracting the mean and dividing by the standard deviation of the closest preceding 'pre' period. For electrical stimulation experiments, the 30 s preceding each stimulation period was used to normalize the subsequent 1 min stim and 1 min post period. For levodopa and saline experiments, the 20 min prior to injection was used to normalize the subsequent 2.5 hr of signal.

## In vivo electrophysiology

A detailed protocol for in vivo physiology can be found here: https://.doi.org/10.17504/protocols.io.5jyl89w69v2w/v1. Briefly, single-unit activity from microwires was recorded using a commutated (Doric) multiplexed 96-channel recording system (CerePlex Direct, Blackrock Microsystems). Spike waveforms were filtered at 154–8800 Hz and digitized at 30 kHz. The experimenter manually set a threshold for storage of electrical events. Spike sorting and single units (SUs) were identified offline by manual sorting into clusters (Offline Sorter, Plexon, RRID:SCR_000012).

Waveform features used for separating units were typically a combination of valley amplitude, the first three principal components (PCs), and/or nonlinear energy. Clusters were classified as single units if they fulfilled the following criteria: (1) <1% of spikes occurred within the refractory period and (2) the cluster was statistically different (p<0.05, MANOVA using the aforementioned features) from the multi- and other single-unit clusters on the same wire.

## Ex vivo slice electrophysiology and imaging

A detailed protocol for slice electrophysiology can be found at https://doi.org/10.17504/protocols.io.6qpvr67rpvmk/v1. To prepare ex vivo slices for whole-cell recordings and GCaMP imaging, mice were deeply anesthetized with IP ketamine-xylazine, transcardially perfused with ice-cold glycerol-based slicing solution, decapitated, and the brain was removed. Glycerol-based slicing solution contained (in mM): 250 glycerol, 2.5 KCl, 1.2 NaH$_2$PO$_4$, 10 HEPES, 21 NaHCO$_3$, 5 glucose, 2 MgCl$_2$, 2 CaCl$_2$. The brain was mounted on a submerged chuck, and sequential 275 mm coronal or sagittal slices were cut on a vibrating microtome (Leica), transferred to a chamber of warm (34°C) carbogenated ACSF

containing (in mM) 125 NaCl, 26 NaHCO$_3$, 2.5 KCl, 1 MgCl$_2$, 2 CaCl$_2$, 1.25 NaH$_2$PO$_4$, 12.5 glucose for 30–60 min, then stored in carbogenated ACSF at room temperature. Each slice was then submerged in a chamber superfused with carbogenated ACSF at 31–33°C for recordings. STN neurons were targeted using differential interference contrast optics in VGlut2-Cre mice on an Olympus BX 51 WIF microscope.

For combined electrophysiology-imaging experiments with GCaMP6s, neurons were patched in the whole-cell current-clamp configuration using borosilicate glass electrodes (3–5 MOhms) filled with potassium gluconate-based internal solution containing (in mM): 113 K-Gluconate, 9 HEPES, 4.5 MgCl$_2$, 14 Tris$_2$-phosphocreatine, 4 Na$_2$-ATP, 0.3 tris-GTP; ~290 mOsm, pH: 7.2–7.25. For opsin validation experiments, neurons were patched in the cell-attached configuration using borosilicate glass electrodes (3–5 MOhms) filled with ACSF. Picrotoxin was added to all external solutions for opsin validation. All recordings were made using a MultiClamp 700B amplifier (Molecular Devices) and digitized with an ITC-18 A/D board (HEKA). Data were acquired using Igor Pro 6.0 software (Wavemetrics, RRID:SCR_000325) and custom acquisition routines (mafPC, courtesy of MA Xu-Friedman; https://www.xufriedman.org/mafpc). Recordings were filtered at 5 kHz and digitized at 10 kHz.

To validate ChR2 function in slice, light pulses were delivered to the slice by a TTL-controlled LED (Olympus), passed through a GFP (473 nm) filter (Chroma) and the 40× immersion objective. LED intensity was adjusted to yield an output of 3 mW at the slice. Light was delivered in 1 min epochs, at 50 Hz, 3 ms pulse width or continuously. Stimulation lasted for 1 min and was preceded and followed by 30 s of recording without stimulation.

For simultaneous electrophysiology and GCaMP6s imaging, current-clamped neurons were stimulated (0.5–1 nA) to elicit action potentials. Stimulation occurred at 10, 20, 40, 50 or 60, and 100 or 120 Hz (100 μs pulse-width); or was delivered as a long single square wave of constant current for 1 min, preceded and followed by 30 s without stimulation. During the duration of each 2 min trial GCaMP fluorescence was either acquired through one-photon or two-photon microscopy. One-photon experiments used a 473 nm light (TTL-controlled LED, Olympus, paired with GFP filter, Chroma) delivered to the slice at <1 mW, with GCaMP6s fluorescence captured using an imaging camera attached to the microscope (QI Retiga Electro). For two-photon microscopy, a two-photon source (Coherent Ultra II) was tuned to 810 nm to identify GCaMP expressing neurons and tuned to 940 nm for calcium imaging. Epi- and transfluorescence signals were captured through a 40×, 0.8 NA objective paired with a 1.4 NA oil immersion condenser (Olympus) to photomultiplier tubes (H10770PA-40 PMTs, Hamamatsu). Data were collected in line scan mode (2–2.4 ms/line, including mirror flyback).

All ex vivo electrical recordings were passed through a 1 Hz high-pass filter to remove slow electrical drift and spikes were extracted using the findpeaks function in Matlab. All ex vivo optical recordings were first collapsed into a one-dimensional fluorescence time series by averaging the fluorescence of pixels within a defined region-of-interest. In one-photon recordings, this signal was further processed by fitting a double exponential and subtracting it to remove effects of signal bleaching.

## Optogenetic manipulations

A detailed protocol for optogenetic manipulations can be found at https://doi.org/10.17504/protocols.io.4r3l2oybxv1y/v1. Prior to optical stimulation experiments, animals were habituated to tethering with custom lightweight patch cables (Precision Fiber Products and ThorLabs) coupled to an optical commutator (Doric Lenses) in the open field for 30 min per day, over 1–2 days. Optical stimulation sessions consisted of five 1 min stimulation periods, each preceded and followed by 1 min of no stimulation, for a total of 11 min. TTL-controlled (Master8, A.M.P.I.) blue laser light (488 nm, 3 mW, Shanghai Laser and Optics Century) was delivered in pulse trains (3 ms, 50 Hz) or continuously. Behavior was rated by an observer blinded to the injected construct (ChR2-eYFP or eYFP).

## Histology and microscopy

A protocol for immunohistochemistry can be found here: https://doi.org/10.17504/protocols.io.14egn7nezv5d/v1. Mice were terminally anesthetized with IP ketamine (200 mg/kg) and xylazine (40 mg/kg). For mice with an implanted STN DBS device or multielectrode array, the site of stimulation or recording was marked with a solid state, direct current Lesion Maker (Ugo Basile). Mice were then transcardially perfused with 4% paraformaldyde (PFA), the brain was dissected from the skull and fixed overnight in 4% PFA, and then was placed in 30% sucrose at 4°C for 2–3 days. Brains were then cut

into 50 µm sagittal sections on a freezing microtome (Leica). To confirm dopamine depletion, tissue was immunostained for tyrosine hydroxylase (TH). Stitched multi-channel fluorescence images were taken on a Nikon 6D conventional widefield microscope at 4–10×, using custom software (UCSF Nikon Imaging Center) to confirm virus expression, fiber placement, and STN DBS placement.

## Group allocation and blinding

The order in which each mouse received electrical stimulation during optical recording experiments was randomized daily, as was the type of stimulation administered. Mouse order and stimulation type was also randomized during optogenetic experiments. For optical manipulations, VGluT-Cre-positive littermates were randomized to eYFP or ChR2-eYFP injection. The experimenter was blinded to experimental group (eYFP vs ChR2-eYFP) during behavioral experiments.

## Inclusion/exclusion criteria

If the number of ipsilesional TH-positive SNc neurons were >5% of the contralateral (unlesioned) side, the animal was excluded from all analyses. Animals that did not show strong virus expression, proper optical fiber placement (within target structure or <0.2 mm above), and/or proper STN DBS device placement (within STN) were excluded from further analysis. If DBS leads developed a short, further DBS experiments were terminated, but levodopa experiments were continued.

A short developed in one mouse in the STN DBS/SNr fiber photometry cohort, such that only levodopa experiments could be performed. We failed to deliver 1 of the 30 DBS parameter combinations in one mouse from the STN DBS/M1 hyperdirect fiber photometry cohort, so no 100 Hz DBS data were included for this mouse. Across all photometry cohorts, 4 mice were excluded due to insufficient GCaMP expression and/or improper targeting of the optical fiber, resulting in no detectable GCaMP signal (4 of 30 mice). Across optical stimulation cohorts, 2 mice were excluded due to insufficient eYFP or ChR2-eYFP expression (2 of 22 optical stimulation mice). One mouse in the STN in vivo electrophysiology cohort (1 of 4) and 1 mouse from the SNr in vivo electrophysiology cohort (1 of 4) were excluded due to a lack of clearly isolated single units.

## Sample size determination, quantification, statistical analysis, and replication

For in vivo fiber photometry experiments, no similar studies had been performed by which to estimate effect size. We performed small pilot experiments in each recorded brain region to determine the mean and standard deviation of GCaMP signals in these regions. Effect sizes were estimated from these pilots, or from similarly designed and published electrophysiological studies. The sample size was calculated using 0.90 power to detect a significant difference, two-sided nonparametric comparisons, and alpha of 0.05. Sample sizes for ex vivo experiments, in vivo electrophysiology, and in vivo optical stimulation were calculated using a similar approach, but based on variance and effect sizes from previous experiments conducted in the lab using similar methods as well as published studies from other laboratories.

All data are expressed as mean ± standard error of the mean (SEM). For all bar graphs for electrical stimulation, optogenetic, and slice experiments, the 'stim' or 'laser' bar was calculated by averaging all 1 min stimulation periods for each trial. The 'pre' and 'post' bars were calculated by averaging the 30 s before and 30 s after each stimulation period, respectively. For all bar graphs involving levodopa or saline, the 'LD' or 'saline' bar was calculated by averaging the 10 min between 30 and 40 min post injection for each trial. The 'pre' and 'post' bars were calculated by averaging the time period between 15 and 5 min before injection and between 125 and 135 min post injection, respectively, for each trial. Correlation between photometry signal and velocity (*Figure 4—figure supplement 1E*) was calculated using *fitlm* in Matlab. Rise time of velocity and calcium signals was calculated as the time it took from the onset of stimulation for the signal to first reach the mean value for that stimulation epoch. Movement starts were defined as events when the mouse's movement velocity changed from less than 0.5 cm/s (maintained for at least 1 s) to more than 2 cm/s. Change in photometry signal or firing rate around movement starts (insets in *Figure 6D* and *Figure 7F–G*) was calculated by subtracting the average fiber signal or firing rate during 1 s preceding movement start from the average fiber signal or firing rate during 1 s following movement start.

All data on which a repeated measures one-way ANOVA (rmANOVA) was performed were tested for normality using a Kolmogorov-Smirnov test. Nonparametric tests (Friedman test, Wilcoxon signed-rank, Wilcoxon rank-sum) were used in all other cases (see *Supplementary file 1*, table 1 for full details). For rmANOVAs and Friedman tests, a Tukey HSD post hoc analysis was applied to correct for multiple comparisons. Data was considered statistically significant for $p < 0.05$.

Multiple cohorts were used in each experiment, and findings were reliably reproduced among all subjects. In vivo optical recordings for each brain region were conducted using at least three cohorts of animals. Optogenetic experiments were conducted with three cohorts of animals. Electrophysiology experiments were conducted with at least two cohorts of animals. M1 lesion experiments were conducted with two cohorts of animals.

## Acknowledgements

The authors would like to acknowledge M McGregor, P Starr, J Ostrem, G Bouvier, M Scanziani, and members of the Nelson and Bender Labs for providing advice and feedback on the manuscript. This work was supported by the NINDS (K08 NS081001, ABN; R01, ABN; F31 NS110329, JSS) and the UCSF Discovery Fellows Program (JSS). This research was also funded in part by Aligning Science Across Parkinson's through the Michael J Fox Foundation for Parkinson's Research (MJFF). For the purpose of open access, the author has applied a CC BY public copyright license to all Author Accepted Manuscripts arising from this submission. ABN is the Richard and Shirley Cahill Endowed Chair in Parkinson's Disease Research.

## Additional information

### Funding

| Funder | Grant reference number | Author |
| --- | --- | --- |
| Aligning Science Across Parkinson's | ASAP-020529 | Alexandra B Nelson |
| National Institutes of Health | F31 NS110329 | Jonathan S Schor |
| National Institutes of Health | K08 NS081001 | Alexandra B Nelson |
| National Institutes of Health | R01NS101354 | Alexandra B Nelson |

The funders had no role in study design, data collection and interpretation, or the decision to submit the work for publication.

### Author contributions

Jonathan S Schor, Conceptualization, Data curation, Formal analysis, Funding acquisition, Investigation, Methodology, Validation, Visualization, Writing - original draft, Writing - review and editing; Isabelle Gonzalez Montalvo, Rea J Brakaj, Jasmine A Stansil, Emily L Twedell, Investigation; Perry WE Spratt, Investigation, Methodology, Visualization; Kevin J Bender, Conceptualization, Investigation, Methodology, Writing - review and editing; Alexandra B Nelson, Conceptualization, Funding acquisition, Investigation, Project administration, Resources, Supervision, Validation, Writing - review and editing

### Author ORCIDs

Jonathan S Schor ⓘ http://orcid.org/0000-0002-2806-8782
Kevin J Bender ⓘ http://orcid.org/0000-0001-7084-1532
Alexandra B Nelson ⓘ http://orcid.org/0000-0002-9305-5662

### Ethics

This study was performed in accordance with the recommendations in the Guide for the Care and Use of Laboratory Animals of the National Institutes of Health. All animal experiments were approved

by the UC San Francisco institutional animal care and use committee (IACUC), protocol # AN189295. Efforts were made throughout to minimize the suffering of animals by use of appropriate anesthetics and analgesics, as well as enrichment and supportive care.

### Decision letter and Author response
Decision letter https://doi.org/10.7554/eLife.75253.sa1
Author response https://doi.org/10.7554/eLife.75253.sa2

---

## Additional files

### Supplementary files
• Supplementary file 1. (a) Full statistical results for the indicated figures. rmANOVA = one-way repeated measures analysis of variance with post hoc Tukey multiple comparisons test. WRS = Wilcoxon rank-sum test (Mann-Whitney U test). FT = Friedman's test with post hoc Tukey multiple comparisons test. WSR = Wilcoxon signed-rank test. (b) Stimulation parameters used in assessing differences between low-, medium-, and high-frequency stimulation. (c) Stimulation parameters used in assessing differences between high and low effect stimulation.

• Transparent reporting form

### Data availability
Source data can be found on Dryad, doi: https://doi.org/10.7272/Q60P0X95.

The following dataset was generated:

| Author(s) | Year | Dataset title | Dataset URL | Database and Identifier |
|---|---|---|---|---|
| Nelson AB | 2022 | Data from: Therapeutic deep brain stimulation disrupts movement-related subthalamic nucleus activity in Parkinsonian mice | http://dx.doi.org/10.7272/dryad.Q60P0X95 | Dryad Digital Repository, 10.7272/dryad.Q60P0X95 |

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
