## [Editor Report]

Using a combination of electrical artifact-free calcium imaging and electrical stimulation, the authors probe the effects of stimulation on the neural dynamics of basal ganglia structures that correlate with motor improvement. The paper would be of interest to neuroscientists and clinician scientists interested in better understanding the mechanism of deep brain stimulation (DBS) in the treatment of Parkinson's disease.

---

## [Decision Letter]

**Decision letter after peer review:**

Thank you for submitting your article "Therapeutic Deep Brain Stimulation Disrupts Subthalamic Nucleus Activity Dynamics in Parkinsonian Mice" for consideration by *eLife*. Your article has been reviewed by 3 peer reviewers, and the evaluation has been overseen by a Reviewing Editor and John Huguenard as the Senior Editor. The following individual involved in review of your submission has agreed to reveal their identity: Dieter Jaeger (Reviewer #3).

Essential revisions:

The reviewers have a good consensus that this paper has many strengths and was thoughtfully done, a few major concerns need to be addressed:

1) it is well-known that clinical DBS employed at least 120 Hz stimulation. But the present study utilized suboptimal frequencies (60 and 100Hz only) to address the mechanism. The loss of correspondence between spiking and calcium at the highest stimulation frequencies produces concern regarding the interpretation of the calcium signals with the "high effect stim" which includes multiple stim protocols with frequency > 100Hz. Added controls with higher frequency stimulation could improve the interpretability of the GCaMP signals for DBS at higher frequencies

2) In addition to establishing baseline firing dynamics in the STN in parkinsonian mice, identification of the movement-related calcium activity in a healthy control mouse would establish whether therapeutic DBS shifts the activity in the direction of healthy STN firing activity (Figure 1B). This could enhance the relevance of the finding that the movement-related neural dynamics in the STN are abolished with DBS.

3) Both Reviewers 1 and 3 suggested employing a miniscope approach to identify the spatial patterns of STN neuronal activities. However, both reviewers also indicated that this would be a whole new study from scratch. Therefore, please take this into consideration when you extend the discussion on the limitation of using fiber photometry.

4) The reviewers also made suggestions on how to enhance the clarity of the paper. In addition, the limitation of fiber photometry should be thoroughly discussed.

*Reviewer #1 (Recommendations for the authors):*

With reference to the comments described in the public review, the authors are recommended to:

1) Replace CGaMP fibre photometry approach by mini-endoscopy technique that could provide single cell level calcium activity. Only using such an approach can provide good enough temporal resolution at single neuron level in reporting firing pattern and synchrony of neurons and their correlation with behavioral effect of DBS while at the same time avoid electrical stimulation artefacts.

2) Report the difference between healthy control and PD mice with respect to calcium activities (preferably via miniscope rather than fibre photometry) in the target areas (STN, SNr, M1) studied, to provide a clear picture of what normal calcium activities in healthy animals are what pathological changes are PD animals.

3) Apply clinically relevant DBS frequency (120Hz or above) in addressing the mechanisms of DBS.

4) Provide videos of the STN stimulation protocol such that readers have a good idea how the animals behave before and after 6-OHDA lesion and under DBS (and preferably in control animals too). The reason for this is that the short (1 min) and robust triggering of movement under DBS gives an impression that the animals were driven to move involuntarily rather than facilitating their natural voluntary movements. If so, the DBS cannot be regarded as therapeutic. Given that only a single parameter was used in this study and other motor abilities are not reported (of which the authors should also include as much as possible), this is considered an important point.

5) In relation to 4) above, what was the effect of removing M1 on the general motor ability of the animal? Would the effect in any way compromise the interpretation of the study of M1 removal in arguing against the antidromic hypothesis?

*Reviewer #2 (Recommendations for the authors):*

In the abstract and introduction, the 3 theories include one as "inhibits STN output" and the other as "disrupts firing dynamics within the STN". The terminology of the latter could be made more specific to avoid overlap between these two. Maybe something like "movement-related activity" instead of "firing dynamics".

The movement-related signal with in vivo electrophysiology and calcium imaging in 1J and 1K is described to be qualitatively similar due to the peak, but the time-course of the recovery from the peak is not addressed. This could affect further interpretability of the duration of changes in the calcium signals.

In making the claim that DBS and levodopa both improve movement, but affect the overall activity differentially, the firing activity response to levodopa is compared to the calcium signal in response to the DBS. This comparison across techniques seems weak, given the differential sensitivity discussed and potentially different mechanisms measured by the two techniques (Figure 2F, J).

M1 lesion experiment, velocity scale reduced to 1cm/s (Figure 5B). Further commentary on the movement status of these lesioned animals.

Further experimental evidence or explanation for the identification of a 50 Hz pulse stim protocol to be the one that mimics the STN dynamics crucial for effective stim response.

The opposing effects of therapeutic electrical and optical DBS on the STN firing rate could be further explained with potential cell-type specific mechanisms of action.

*Reviewer #3 (Recommendations for the authors):*

Generally the paper is very well written, and the data presented clearly. Figures are of excellent quality.

---

## [Author Response]

Essential revisions:The reviewers have a good consensus that this paper has many strengths and was thoughtfully done, a few major concerns need to be addressed:(1a) It is well-known that clinical DBS employed at least 120 Hz stimulation. But the present study utilized suboptimal frequencies (60 and 100Hz only) to address the mechanism.

We thank the reviewer for bringing up these important points, which may not have been clear from the manuscript. First of all, though DBS has traditionally been targeted to higher frequencies (120-200 Hz), accumulating clinical evidence suggests that lower frequencies can be quite effective, particularly if the pulse width and/or amplitude are increased to compensate for the lower frequency of stimulation (Blumenfeld et al., 2017; Conway et al., 2021; Khoo et al., 2014; Moreau et al., 2008; Xie et al., 2015). Some studies suggest lower frequency stimulation (eg 60 Hz) might be particularly helpful for freezing of gait and other “axial” symptoms. For this reason, we think lower DBS frequencies are clinically relevant.

In the mouse model of STN DBS for Parkinson’s Disease, we previously showed that when pulse width and amplitude are fixed, therapeutic increases in movement velocity tended to increase with stimulation frequency from 15-120 Hz, but that low frequency stimulation could be as effective when paired with higher pulse width and/or amplitude (Schor and Nelson, 2019). In this same manuscript, we re-plotted previously published data on finger tapping in Parkinson’s Disease patients during DBS (Moro et al., 2002), and saw a similar effect. Together, these results suggest that DBS at moderate frequencies, such as 50-60 or 100 Hz, is clinically relevant in PD patients and therapeutic in the mouse model.

Based on our previous findings that moderate (60-100 Hz) frequency DBS was effective in the mouse model, and our plan to use optical tools, such as ChR2, which is known from ex vivo studies to have reduced spike fidelity starting around 60 Hz, but certainly above 100 Hz, we decided to highlight 50-60 (optical and electrical experiments, respectively) and 100 Hz DBS in the main figures. However, in the revised manuscript, we have included experiments using STN DBS with 12 different stimulation parameter sets, over a range of frequencies, while holding amplitude and pulse width fixed, including the higher frequencies that the reviewer points out are more commonly used in PD patients (Supplementary File 1b). As in our original paper on the STN DBS model, we found that many parameter combinations were effective, including some at lower frequencies. We have now included those data in new supplements: Figure 2—figure supplement 2D, Figure 3—figure supplement 1D, and Figure 4—figure supplement 1E. We have separated trials into three groups by DBS frequency: “Low” (5-40Hz), “Medium” (60-100 Hz) and “High” (120-180 Hz) for the analysis. In the supplementary figures, the behavioral effect of DBS on movement velocity is shown for each of these groups alongside the physiological effects on STN (Figure 2—figure supplement 2D), SNr (Figure 3—figure supplement 1D), and hyperdirect pathway activity (Figure 4—figure supplement 1D). The effects are qualitatively and quantitatively quite similar for “Medium” and “High” frequency stimulation. “Low” frequency stimulation produced clear, but more modest effects on movement velocity and neural activity, consistent with our previously published work in this model.

(1b) The loss of correspondence between spiking and calcium at the highest stimulation frequencies produces concern regarding the interpretation of the calcium signals with the "high effect stim" which includes multiple stim protocols with frequency > 100Hz. Added controls with higher frequency stimulation could improve the interpretability of the GCaMP signals for DBS at higher frequencies.

The reviewer brings up an excellent point. First, we want to clarify that the ex vivo GCaMP/whole-cell electrophysiology experiments performed in Figure 1A-G were meant to validate that STN neuron GCaMP fluorescence increases with higher rates of STN neuron spiking activity, rather than directly test how GCaMP fluorescence reflects different stimulation frequencies of DBS. In the ex vivo experiments, intracellular current pulses were applied to drive STN neurons at specific firing rates. These experiments did not deliver extracellular pulses like DBS, which presumably recruits many different circuit elements as well as cellular compartments. We do not know the frequencies of spiking that are elicited by different frequencies of electrical stimulation in vivo, as we have performed optical recordings in vivo. It is quite possible that 100 Hz electrical stimulation does not evoke 100 Hz spiking in vivo, and so on. We apologize for the confusion and have tried to clarify in the Results text the intent and methodology for the ex vivo experiments.

However, the reviewer makes an excellent suggestion to test the relationship of STN GCaMP fluorescence to STN spiking at higher firing rates. We have now performed additional combination ex vivo whole-cell electrophysiology/calcium imaging experiments, included in Figure 1E, G. First, we found that individual STN neurons can be driven with high frequency intracellular current injections to fire at correspondingly high firing rates (up to 200 Hz, Figure 1E inset). Second, in this group of neurons driven to fire at very high firing rates, there is increasing GCaMP fluorescence with higher firing rates up to 100 Hz (consistent with the data shown in the original manuscript), but in most cells there is a more modest increase in STN neuron GCaMP fluorescence between 100-200 Hz spiking.

2) In addition to establishing baseline firing dynamics in the STN in parkinsonian mice, identification of the movement-related calcium activity in a healthy control mouse would establish whether therapeutic DBS shifts the activity in the direction of healthy STN firing activity (Figure 1B). This could enhance the relevance of the finding that the movement-related neural dynamics in the STN are abolished with DBS.

In response to the reviewer’s suggestion, we have performed additional in vivo STN GCaMP fiber photometry experiments in healthy mice (Figure 1J, M; N=5). These animals, as expected, move at higher average velocities than do 6-OHDA treated animals. In healthy mice, as in the 6-OHDA-treated mice, we observed an increase in STN calcium signal (Figure 1M, top) when aligned to movement starts (Figure 1M, bottom). This observation is in line with several published papers, in other species and with other methodologies, showing movement-related increases in STN activity in healthy animals (eg Cheruel et al., 1996; Wichmann et al., 1994).

The reviewer was interested in the possibility that DBS might shift activity in the direction of “healthy” STN activity. For this possibility to be considered with calcium signals (as the reviewer suggested), there must be a difference between movement-related STN calcium signals in healthy vs 6-OHDA treated mice. Comparing the average normalized STN GCaMP fiber photometry signals in Figure 1L and 1M qualitatively, the 6-OHDA group shows a dip in STN activity just before movement starts, which was not seen in healthy animals. The duration of the increase in STN signal after movement starts also appears to be longer in healthy animals. However, we are reluctant to make any direct comparisons between these groups, for several reasons. As the absolute value of the GCaMP signal varies from mouse to mouse, based on GCaMP viral expression, precise fiber placement, among other factors, it is challenging, even with normalization, to make direct comparisons across mice. In addition, though we have aligned to movement starts in both healthy and 6-OHDA treated animals, movement kinematics are likely to differ between healthy and 6-OHDA-treated animals. Even the overall velocity trajectory is quite different, as seen in the bottom panels of Figure 1L and 1M. This could contribute to the different shapes of the fiber photometry signal, as well. We suspect that in future work, to determine how the movement-related STN activity might differ between healthy and parkinsonian animals, it would be best to perform STN single-unit electrophysiology, in which an absolute value (firing rate/pattern) can be determined and permits cross-animal comparisons. The interpretation of such findings would also benefit from comparison of similar movement trajectories in the two groups, either through a more fine-grained analysis of spontaneous movement, or even use of a trained behavior. Based on the findings we have now included in the paper, however, we do not believe we can make a strong argument that STN DBS is restoring a “healthy” pattern of movement-related STN activation, just that STN DBS suppresses movement-related STN activity as measured by GCaMP.

3) Both Reviewers 1 and 3 suggested employing a miniscope approach to identify the spatial patterns of STN neuronal activities. However, both reviewers also indicated that this would be a whole new study from scratch. Therefore, please take this into consideration when you extend the discussion on the limitation of using fiber photometry.

We appreciate the reviewers’ suggestions regarding the use of miniscope calcium imaging to address questions related to those we examined. We thank the reviewers for agreeing that though such an approach provides the opportunity to examine individual cellular Ca signals and is likely to be quite interesting, such experiments would entail a brand new study and would not be feasible to develop in our lab on a reasonable timescale. We have, in keeping with many of the suggestions by reviewers, devoted additional space to discussing the limitations of fiber photometry, above and beyond those associated with GCaMP, and the potential benefits of future single-cell resolution calcium imaging in conjunction with DBS.

4) The reviewers also made suggestions on how to enhance the clarity of the paper. In addition, the limitation of fiber photometry should be thoroughly discussed.

As noted above, in the revised manuscript we have devoted more space in the Discussion to the caveats of GCaMP and fiber photometry. We appreciate the reviewers’ additional suggestions and they are noted in response to each specific critique/suggestion below.

Reviewer #1 (Recommendations for the authors):With reference to the comments described in the public review, the authors are recommended to:1) Replace CGaMP fibre photometry approach by mini-endoscopy technique that could provide single cell level calcium activity. Only using such an approach can provide good enough temporal resolution at single neuron level in reporting firing pattern and synchrony of neurons and their correlation with behavioral effect of DBS while at the same time avoid electrical stimulation artefacts.

We agree with the Reviewer that single-cell calcium imaging, in addition to its ability to resolve individual cells, could provide additional information about the spatial and temporal features of STN calcium responses to electrical DBS. We appreciate this suggestion, but as noted above, repeating all the experiments with a miniscope approach would essentially entail a brand-new study. It would require optimizing this technique with electrical DBS, developing computational analysis of these signals, and running scores of additional animals and analyzing their data (we are not currently performing miniscope recordings in our lab). Even if a new trainee with expertise in this area were available, we estimate it would take us 2+ years to gather the requisite data. Finally, current GCaMP indicators combined with miniscope techniques in freely moving animals have single-cell resolution, and are likely to reduce or eliminate some of the confounds associated with collecting calcium signals from the entire cell https://www.biorxiv.org/content/10.1101/2021.01.20.427525v1, but to our knowledge still cannot achieve single-spike resolution, particularly in a tonically active nucleus such as the STN. This approach would certainly offer higher temporal resolution than fiber photometry, but still not at the level of spikes with current technology. We hope that in the future, improvements in calcium and/or voltage indicators usable in deep structures like the STN in freely moving animals will permit a closer approximation to the information in electrophysiology, as this would truly get to some of the Reviewer’s critical questions regarding neural patterning and synchrony.

2) Report the difference between healthy control and PD mice with respect to calcium activities (preferably via miniscope rather than fibre photometry) in the target areas (STN, SNr, M1) studied, to provide a clear picture of what normal calcium activities in healthy animals are what pathological changes are PD animals.

We share the reviewer’s interest in the differences between healthy and 6-OHDA treated animals in the target areas. There are limitations in comparing the absolute value of calcium signals across animals, but we have now performed STN GCaMP fiber photometry in healthy animals and see a rise in normalized GCaMP signals at movement onset (Figure 1M) roughly similar to that in 6-OHDA treated mice. This observation mirrors the increase in firing rate (as measured with electrophysiology) at movement initiation we observed in parkinsonian mice (Figure 1K) and has been reported in other species (Cheruel et al., 1996; Wichmann et al., 1994).

To the specific question of what is different in the physiology of basal ganglia neurons between healthy and parkinsonian states (independent of DBS), we think that single-unit electrophysiology is likely the most sensitive and interpretable approach, as it gives single-cell and spike-level resolution, and provides an absolute activity level (firing rate). There are a large number of careful electrophysiological studies in nonhuman primates, rats, and increasingly, mice, that have provided these comparisons in the STN, SNr and M1, though mostly with the one-cell-at-a-time approach. In the future, we anticipate high density electrophysiological probes can give an even better sense of the differences in single-cell and ensemble basal ganglia activity between healthy and parkinsonian animals.

3) Apply clinically relevant DBS frequency (120Hz or above) in addressing the mechanisms of DBS.

The Reviewer makes an excellent point, in that most clinical DBS in Parkinson’s Disease is performed at high frequencies. However, there is accumulating evidence that lower frequencies of DBS can be clinically effective (see comments under Essential Revisions/Critique 1a).

In response to the Reviewer’s suggestion, we have now included data from STN, SNr, M1 GCaMP fiber photometry in parkinsonian mice with DBS at 12 different parameter combinations, ranging from 5 Hz to 180 Hz in stimulation frequency. We observed qualitatively similar findings in terms of fiber photometry signal across these combinations. We have now included analyses of different frequencies in this range with fixed pulse width and amplitude (Figure 2—figure supplement 2D, Figure 3—figure supplement 1D, and Figure 4—figure supplement 1E). See details under Essential Revisions/Critique 1a.

Though the frequency of stimulation in DBS does not necessarily match the frequency of STN firing, we have also performed additional ex vivo GCaMP validation experiments, up to a firing rate of 200 Hz (Figure 1E, G, insets). See details under Essential Revisions/Critique 1b.

4) Provide videos of the STN stimulation protocol such that readers have a good idea how the animals behave before and after 6-OHDA lesion and under DBS (and preferably in control animals too). The reason for this is that the short (1 min) and robust triggering of movement under DBS gives an impression that the animals were driven to move involuntarily rather than facilitating their natural voluntary movements. If so, the DBS cannot be regarded as therapeutic. Given that only a single parameter was used in this study and other motor abilities are not reported (of which the authors should also include as much as possible), this is considered an important point.

We thank the reviewer for this comment and suggestion. We agree very large increases in movement could be pathological. Therapeutic DBS in human patients doesn’t make movement obligatory, but does increase spontaneous movement, and makes it more fluid and normal in speed when voluntarily initiated. Though we cannot easily distinguish voluntary versus involuntary movements in mice, we do observe that in a given mouse, during a single session of DBS (see Author response image 1) some stimulation epochs are associated with substantial increases in movement velocity, others with modest increases (upper trace, arrowhead), and others with essentially no change (bottom trace, arrowheads). If a mouse wants to sit, DBS doesn’t seem to prevent that. [The dotted lines represent 2 SD above the mean velocity for that session.] We do not have paired videos from the same mouse before 6-OHDA treatment, after 6-OHDA treatment, and during DBS. We have collected videos shot from above the mouse during DBS sessions, including ON and OFF stimulation. However, we point the Reviewer to a video (shot from the side) associated with our previously published manuscript characterizing the mouse model of STN DBS for PD, which includes baseline, ON, and OFF periods. We used the identical DBS approach as used in this manuscript. https://www.jci.org/articles/view/122390/sd/2

**Author response image 1. sa2fig1:** 

5) In relation to 4) above, what was the effect of removing M1 on the general motor ability of the animal? Would the effect in any way compromise the interpretation of the study of M1 removal in arguing against the antidromic hypothesis?

The reviewer brings up an excellent point. Is motor function overall suppressed in parkinsonian animals after M1 lesions? Our primary metric, average locomotor velocity, ranged from 0.5-1 cm/sec across parkinsonian imaging cohorts shown in Figures 2 and 3, and baseline velocity was on the lower end of this range (approximately 0.5 cm/sec) in parkinsonian animals after M1 lesion (Figure 4). Despite baseline velocities on the lower end of what was seen in other cohorts, M1-lesioned mice showed improvement with STN DBS, supporting the idea that the hyperdirect pathway is not the only or primary pathway involved in the therapeutic effects of STN DBS in parkinsonian mice.

We did not perform assays of fine motor control in this study, so cannot address whether electrical STN DBS remediates fine motor control in 6-OHDA treated mice, nor whether our M1 lesions produced deficits in fine motor control. STN DBS in PD patients produces a number of benefits, including improvements in gross and fine motor control. First, it is possible that though M1 is not required for the gross motor benefits we observed in 6-OHDA treated mice, it would be required for improvements (if present) in fine motor assays. Second, the mouse brain is not the same as the human brain, and the 6-OHDA model is the not the same as Parkinson’s Disease; it remains quite possible that the hyperdirect pathway plays a more significant part in the therapeutic benefits of STN DBS in PD patients than in the mouse model. We appreciate the Reviewer raising these issues and have now expanded our discussion on the hyperdirect pathway and limitations in the interpretation of our results.

Reviewer #2 (Recommendations for the authors):In the abstract and introduction, the 3 theories include one as "inhibits STN output" and the other as "disrupts firing dynamics within the STN". The terminology of the latter could be made more specific to avoid overlap between these two. Maybe something like "movement-related activity" instead of "firing dynamics".

We appreciate the Reviewer’s suggestion and agree that “firing dynamics” in this context is a little confusing, and have changed the Title to “Therapeutic Deep Brain Stimulation Disrupts Movement-Related Subthalamic Nucleus Activity in Parkinsonian Mice”. It is challenging to come up with a unifying phrase to encompass theories that STN DBS changes “dynamics”, as published work on this topic includes many papers using local field potential (LFP) as a readout, and mostly without aligning neural activity to motor output. However, we understand the concern, and have attempted to clarify the way we describe our own findings, which are specific to movement-related STN activity, as the Reviewer notes.

The movement-related signal with in vivo electrophysiology and calcium imaging in 1J and 1K is described to be qualitatively similar due to the peak, but the time-course of the recovery from the peak is not addressed. This could affect further interpretability of the duration of changes in the calcium signals.

We appreciate the reviewer’s comment and have expanded our discussion of the temporal features of GCaMP/fiber photometry versus electrophysiological signals, as well as the movement-aligned increase in STN activity, in the Discussion. The decay of the movement-related increases in firing rate and calcium signal in (revised) Figures 1K and 1L are relatively similar (reaching baseline around 3 seconds later). However, as we now highlight in the discussion, recent work in the striatum using simultaneous multi-unit electrophysiology and GCaMP fiber photometry indicate that photometry signals can outlive spiking by a considerable period, perhaps due to calcium signals derived from the dendrites https://www.biorxiv.org/content/10.1101/2021.01.20.427525v1. This type of experiment has not been performed in the STN, but a similar phenomenon might be expected in most, if not all brain areas due to calcium influx in the dendritic arbors as a result of backpropagation.

In making the claim that DBS and levodopa both improve movement, but affect the overall activity differentially, the firing activity response to levodopa is compared to the calcium signal in response to the DBS. This comparison across techniques seems weak, given the differential sensitivity discussed and potentially different mechanisms measured by the two techniques (Figure 2F, J).

At the time we designed our experiments, there were no similar experiments using in vivo calcium imaging in structures like the STN and SNr in parkinsonian mice, but there were numerous similar studies using single-unit electrophysiological recordings in parkinsonian nonhuman primates and rodents. We felt that given the very well-established effects of levodopa on firing rate/pattern in different basal ganglia nuclei, it would be helpful to show how GCaMP signals changed in response to this extensively studied manipulation. We reasoned the levodopa experiments would provide an additional layer of validation of the technique. However, we agree with the Reviewer that DBS and levodopa are likely to have different impacts on basal ganglia activity, and to relieve parkinsonian symptoms in different ways. In the revised manuscript, we have tried to clarify what can and cannot be compared between these two interventions.

M1 lesion experiment, velocity scale reduced to 1cm/s (Figure 5B). Further commentary on the movement status of these lesioned animals.

See Reviewer 1/comment 5 above.

Further experimental evidence or explanation for the identification of a 50 Hz pulse stim protocol to be the one that mimics the STN dynamics crucial for effective stim response.

We thank the reviewer for this comment. We chose 50 Hz because this was likely to be close to the ceiling light stimulation frequency that could be used with ChR2, based on ex vivo studies (see Figure 7—figure supplement 1D-H), but was relatively close to a highly effective electrical DBS frequency based on our previous work. We were not trying to achieve a specific STN firing rate or pattern with this frequency. We suspect there is nothing particularly special about 50 Hz, and indeed 50 Hz ChR2 stimulation achieves a wide variety of actual firing rates in STN neurons in vivo (Figure 7E, 7C).

The opposing effects of therapeutic electrical and optical DBS on the STN firing rate could be further explained with potential cell-type specific mechanisms of action.

We thank the reviewer for this comment. To clarify, we cannot measure the firing rate of STN neurons with any GCaMP methodology, and moreover fiber photometry reads out calcium signals across a population of neurons, including both their somata and dendrites. Despite some nonlinearities, based on ex vivo and in vivo validation experiments (Figure 1) we suspect that in some ranges of firing, there is a correlation between the two methodologies. The reviewer brings up an excellent point that we found electrical DBS tended to increase STN activity (as assayed with fiber photometry) while optical DBS tended (on average) to decrease STN activity. We interpret this to mean that the average rate of activity is not the unifying mechanism of action. Indeed, as the reviewer notes, these stimulation methods would be expected to recruit different circuit elements. Electrical stimulation preferentially recruits axons over dendrites and cell bodies, for example, and thus when applied to the STN would be expected to activate incoming and outgoing axons, as well as fibers of passage. In turn, such recruitment of axons could cause antidromic spiking in STN input regions (M1 cortex, GPe, etc), orthodromic activation of STN targets (SNr, GPe, GPi, etc), or effects in areas connected to the fibers of passage. Optical stimulation of ChR2-expressing glutamatergic STN cells, on the other hand, would be expected to specifically modulate the activity of cell bodies (as we have validated here with single-unit electrophysiology). By virtue of action potential propagation and synaptic release, optical stimulation would also be expected to modulate activity in the monosynaptic targets of glutamatergic STN neurons (SNr, GPe, GPi, etc), but should not (directly at least) regulate the activity of other regions sending inputs into or through the STN region. The differences in STN activity under these two distinct stimulation methods, thus, may reflect the differences in the cellular elements they recruit.